# Learning Structured Representations with Hyperbolic Embeddings

**Aditya Sinha**[*][†]
University of Illinois, Urbana-Champaign
as146@illinois.edu

**Siqi Zeng**[*]
University of Illinois, Urbana-Champaign
siqi6@illinois.edu

**Makoto Yamada**
Okinawa Institute of Science and Technology
makoto.yamada@oist.jp

**Han Zhao**
University of Illinois, Urbana-Champaign
hanzhao@illinois.edu

## Abstract

Most real-world datasets consist of a natural hierarchy between classes or an inherent label structure that is either already available or can be constructed cheaply. However, most existing representation learning methods ignore this hierarchy, treating labels as permutation invariant. Recent work [104] proposes using this structured information explicitly, but the use of Euclidean distance may distort the underlying semantic context [8]. In this work, motivated by the advantage of hyperbolic spaces in modeling hierarchical relationships, we propose a novel approach HypStructure: a Hyperbolic Structured regularization approach to accurately embed the label hierarchy into the learned representations. HypStructure is a simple-yet-effective regularizer that consists of a hyperbolic tree-based representation loss along with a centering loss. It can be combined with any standard task loss to learn *hierarchy-informed* features. Extensive experiments on several large-scale vision benchmarks demonstrate the efficacy of HypStructure in reducing distortion and boosting generalization performance, especially under low-dimensional scenarios. For a better understanding of structured representation, we perform an eigenvalue analysis that links the representation geometry to improved Out-of-Distribution (OOD) detection performance seen empirically. The code is available at https://github.com/uiuctml/HypStructure.

## 1 Introduction

Real-world datasets, such as ImageNet [72] and CIFAR [45], often exhibit a natural hierarchy or an inherent label structure that describes a structured relationship between different classes in the data. In the absence of an existing hierarchy, it is often possible to cheaply construct or infer this hierarchy from the label space directly [64]. However, the majority of existing representation learning methods [43, 7, 95, 29, 87, 33, 27, 39] treat the labels as permutation invariant, ignoring this semantically-rich hierarchical label information. Recently, Zeng et al. [104] offer a promising approach to embed the tree-hierarchy explicitly in representation learning using a tree-metric-based regularizer, leading to improvements in generalization performance. The approach uses a computation of shortest paths between two classes in the tree hierarchy to enforce the same structure in the feature space, by means of a **Co**phenetic **C**orrelation **C**oefficient (CPCC) [79] based regularizer. However, their approach uses the $\ell_2$ distance in the Euclidean space, distorting the parent-child representations in the hierarchy [70, 50] owing to the bounded dimensionality of the Euclidean space [8].

---

[*]Authors contributed equally.
[†]Now at Netflix Inc.

38th Conference on Neural Information Processing Systems (NeurIPS 2024).

Hyperbolic geometry has recently gained growing interest in the field of representation learning [66, 67]. Hyperbolic spaces can be viewed as the continuous analog of a tree, allowing for embedding tree-like data in finite dimensions with minimal distortion [44, 73, 75, 24]. Unlike Euclidean spaces with zero curvature and spherical spaces with positive curvature, the hyperbolic spaces have negative curvature enabling the length to grow exponentially with its radius. Owing to these advantages, hyperbolic geometry has been used for various applications such as natural language processing [52, 73, 16], image classification [40, 103, 18], object detection [46, 21], action retrieval [55], and hierarchical clustering [100]. However, the aim of using hyperbolic geometry in these approaches is often to *implicitly* leverage the hierarchical nature of the data.

In this work, given a label hierarchy, we argue that accurately and *explicitly* embedding the hierarchical information into the representation space has several benefits, and for this purpose, we propose `HypStructure`, a hyperbolic label-structure based regularization approach that extends the proposed methodology in [104] for semantically structured learning in the hyperbolic space. `HypStructure` can be easily combined with any standard task loss for optimization, and enables the learning of discriminative and *hierarchy-informed* features. In summary, our contributions are as follows:

- We propose `HypStructure` and demonstrate its effectiveness in the supervised hierarchical classification tasks on three real-world vision benchmark datasets, and show that our proposed approach is effective in both training from scratch, or fine-tuning if there are resource constraints.

- We qualitatively and quantitatively assess the nature of the learned representations and demonstrate that along with the performance gains, using `HypStructure` as a regularizer leads to more interpretable as well as tree-like representations as a side benefit. The low-dimensional representative capacity of hyperbolic geometry is well-known [6], and interestingly, we observe that training with `HypStructure` allows for learning extremely low-dimensional representations with distortion values lower than even their corresponding high-dimensional Euclidean counterparts.

- We argue that representations learned with an underlying hierarchical structure are beneficial not only for the in-distribution (ID) classification tasks but also for Out-of-distribution (OOD) detection tasks. We empirically demonstrate that learning ID representations with `HypStructure` leads to improved OOD detection on 9 real-world OOD datasets without sacrificing ID accuracy [106].

- Inspired by the improvements in OOD detection, we provide a formal analysis of the eigenspectrum of the in-distribution *hierarchy-informed* features learned with CPCC-style structured regularization methods, thus leading to a better understanding of the behavior of structured representations in general.

## 2 Preliminaries

In this section, we first provide a background of structured representation learning and then discuss the limited representation capacity of the Euclidean space for hierarchical information, which serves as the primary motivation for our work.

### 2.1 Background

Structured representation learning [104] breaks the permutation invariance of flat representation learning by incorporating a hierarchical regularization term with a standard classification loss. The regularization term is specifically designed to enforce class-conditioned grouping or partitioning in the feature space, based on a given hierarchy.

More specifically, given a weighted tree $\mathcal{T} = (V, E, e)$ with vertices $V$, edges $E$ and edge weights $e$, let us compute a tree metric $d_{\mathcal{T}}$ for any pair of nodes $v, v' \in V$, as the weighted length of the shortest path in $\mathcal{T}$ between $v$ and $v'$. For a real world dataset $\mathcal{D} = \{(\boldsymbol{x}_i, y_i)\}_{i=1}^N$, we can specify a *label tree* $\mathcal{T}$ where a node $v_i \in V$, $v_i$ corresponds to a subset of classes, and $\mathcal{D}_i \subseteq \mathcal{D}$ denote the subset of data points with class label $v_i$. We denote dataset distance between $\mathcal{D}_i$ and $\mathcal{D}_j$ as $\rho(v_i, v_j) = d(\mathcal{D}_i, \mathcal{D}_j)$, where $d(\cdot, \cdot)$ is any distance metric in the feature space, varied by design.

With a collection of tree metric $d_{\mathcal{T}}$ and dataset distances $\rho$, we can use the Cophenetic Correlation Coefficient (CPCC) [79], inherently a Pearson's correlation coefficient, to evaluate the correspondence between the nodes of the tree, and the features in the representation space. Let $\overline{d_{\mathcal{T}}}, \overline{\rho}$ denote the mean of the collection of distances, then CPCC is defined as

$$\text{CPCC}(d_{\mathcal{T}}, \rho) := \frac{\sum_{i<j}(d_{\mathcal{T}}(v_i, v_j) - \overline{d_{\mathcal{T}}})(\rho(v_i, v_j) - \overline{\rho})}{(\sum_{i<j}(d_{\mathcal{T}}(v_i, v_j) - \overline{d_{\mathcal{T}}})^2)^{1/2}(\sum_{i<j}(\rho(v_i, v_j) - \overline{\rho})^2)^{1/2}}. \tag{1}$$

For the supervised classification task, we consider the training set $\mathcal{D}_{\text{tr}}^{\text{in}} = \{(\boldsymbol{x}_i, y_i)\}_{i=1}^N$ and we aim to learn the network parameter $\theta$ for a feature encoder $f_\theta : \mathcal{X} \to \mathcal{Z}$, where $\mathcal{Z} \subseteq \mathbb{R}^d$ denotes the representation/feature space. For structured representation learning, the feature encoder is usually followed by a classifier $g_w$, and the parameters $\theta, w$ are learnt by minimizing $\mathcal{L}$ along with a standard *flat* (non-hierarchical) classification loss, for instance, Cross-Entropy (CE) or Supervised Contrastive (SupCon) [39] loss, with the structured regularization term as:

$$\mathcal{L}(\mathcal{D}) = \sum_{(\boldsymbol{x},y)\in\mathcal{D}} \ell_{\text{Flat}}(\boldsymbol{x}, y, \theta, w) - \alpha \cdot \text{CPCC}(d_{\mathcal{T}}, \rho). \tag{2}$$

Using a composite objective as defined in Equation (2), we can enforce the distance relationship between a pair of representations in the feature space, to behave similarly to the tree metric between the same vertices. For instance, consider a simple label tree with a root node, a coarse level, and a fine level, where subsets of fine classes share the same coarse parent. For this hierarchy, we would expect the fine classes of the same parents (e.g., *apple* and *banana* are *fruits*) to have closer representations in the feature space, whereas fine classes with different coarse parents (e.g., an *apple* is a *fruit* and a *tulip* is a *flower*) should be further apart. The learned structure-informed representations reflect these hierarchical relationships and lead to interpretable features with better generalization [104].

## 2.2 $\ell_2$-CPCC

Equation (1) offers the flexibility of designing a metric to measure the similarity between two data subsets, and [104] define the *Euclidean dataset distance* as $\rho_{\ell_2}(\mathcal{D}_i, \mathcal{D}_j) := \|\frac{1}{|\mathcal{D}_i|} \sum_{\boldsymbol{x}\in\mathcal{D}_i} f(\boldsymbol{x}) - \frac{1}{|\mathcal{D}_j|} \sum_{\boldsymbol{x}'\in\mathcal{D}_j} f(\boldsymbol{x}')\|_2$. The distance between datasets is thus the $\ell_2$ distance between two Euclidean *centroids* of their class-conditioned representations, which is unsuitable for modeling *tree-like* data [8]. Additionally, this regularization approach in [104] is applied only to the leaf nodes of $\mathcal{T}$ for efficiency. However, this leaf-only formulation of the CPCC offers an *approximation* of the structured information, since the distance between non-leaf nodes is not restricted explicitly by the regularization. This approximation, therefore, leads to a loss of information contained in the original hierarchy $\mathcal{T}$. Actually, it is impossible to embed $d_{\mathcal{T}}$ into $\ell_2$ exactly. Or more formally, there exists no bijection $\varphi$ such that $d_{\mathcal{T}}(\varphi(\boldsymbol{z}_i), \varphi(\boldsymbol{z}_j)) = \|\boldsymbol{z}_i - \boldsymbol{z}_j\|_2$ irrespective of how large the feature dimension $d$ is. We provide two such examples for a toy label tree in Figure 1, below.

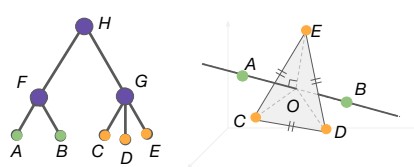

Figure 1: (left) An unweighted label tree with two coarse nodes: $F, G$. $F$ contains two fine classes $A, B$ and $G$ contains three fine classes $C, D, E$. We cannot embed this in $\ell_2$ exactly (right).

**Example 1.** We intend to embed all nodes in $\mathcal{T}$, including purple internal nodes. Notice that $G, C, D, E$ is a star graph centered at $G$. Since $CG = DG = 1, CD = 2$, by triangle inequality $C, D, G$ must be on the same line where $G$ is the center of $CD$. Similarly, $G$ must be at the center of $DE$. Hence, the location of $E$ must be at $C$, which contradicts the uniqueness of all nodes in $\mathcal{T}$.

**Example 2.** As an easier problem, let us only embed leaf nodes into the Euclidean space as shown in Figure 1. Since $CD = DE = CE = 2$, they must be on a plane with an equilateral triangle $\triangle_{CDE}$ in Euclidean geometry. Then all the green classes have the same distance 4 to each yellow class. Therefore, $A, B$ must be on the line perpendicular to $\triangle_{CDE}$ and intersecting the plane with $O$, which is the barycenter of $\triangle_{CDE}$. Due to the uniqueness and symmetry of $A, B$, we must have $AO = BO = 1$ to satisfy $AB = 2$. $AO = 1, OE = \frac{2\sqrt{3}}{3}, AE = 4$ which contradicts the Pythagorean Theorem.

Since we cannot embed an arbitrary tree $\mathcal{T}$ into $\ell_2$ without distortion, it would also affect the optimization of the $\ell_2$-CPCC in a classification problem, where the tree weights encode knowledge

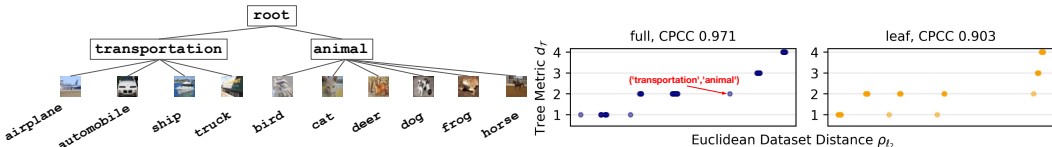

Figure 2: Using $\ell_2$-CPCC for structured representation on CIFAR10. CIFAR10 hierarchy (left) has a three level structure with 13 vertices. For a 512-dimensional embedding, we apply $\ell_2$-CPCC either for the full tree (middle) or the leaf nodes only (right) and plot the ground truth tree metric against pairwise Euclidean centroid distances of the learnt representation. The optimal train CPCC is 1.

of class similarity. To verify our claims, we consider the optimization of 512-dimensional $\ell_2$-CPCC structured representations for CIFAR10 [45]. The CIFAR10 dataset consists of a small label hierarchy as shown in Figure 2 (left). The optimal CPCC is achieved when each tree metric value corresponds to a single $\rho_{\ell_2}$. However, in Figure 2 (right), even with an optimization of the $\ell_2$-CPCC loss for the entire tree, we observe a sub-optimal train CPCC less than 1, where the distance between two coarse nodes, *transportation* and *animal*, is far away from the desired solution. Furthermore, optimization of the CPCC loss for only the leaf nodes, leads to an even larger distortion of the tree metrics.

## 3 Methodology

Hyperbolic spaces are more suitable for embedding hierarchical relationships with low distortion [75] and low dimensions. Hence, motivated by the aforementioned challenges, we propose a **Hyp**erbolic **Structure**d regularizer for *hierarchy-informed* representation learning. We begin with the basics of hyperbolic geometry, followed by the detailed methodology of our proposed approach.

### 3.1 Hyperbolic Geometry

Hyperbolic spaces are non-Euclidean spaces with negative curvature where given a fixed point and a line, there exist infinitely many parallel lines that can pass through this point. There are several commonly used isometric hyperbolic models [4]. For this work, we mainly use the Poincaré Ball model.

**Definition 3.1** (Manifold). A *manifold* $\mathcal{M}$ is a set of points $\boldsymbol{z}$ that are locally Euclidean. Every point $\boldsymbol{z}$ of the manifold $\mathcal{M}$ is attached to a *tangent space* $\mathcal{T}_{\boldsymbol{z}}\mathcal{M}$, which is a vector space over the reals of the same dimensionality as $\mathcal{M}$ that contain all the possible directions that can tangentially pass through $\boldsymbol{z}$.

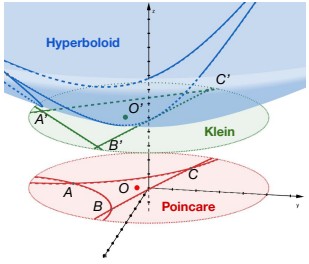

Figure 3: Lines on different models for 2-dimensional hyperbolic space.

**Definition 3.2** (Poincaré Ball Model). Given $c$ as a constant, the Poincaré ball model $(\mathbb{B}_c^d, \mathfrak{g}_B)$ is defined by a manifold of an open ball $\mathbb{B}_c^d = \{\boldsymbol{z} \in \mathbb{R}^d : c\|\boldsymbol{z}\|^2 < 1\}$ and metric tensor $\mathfrak{g}_B$ that defines an inner product of $\mathcal{T}_{\boldsymbol{z}}\mathbb{B}_c^d$. The model is equipped with the distance [88] as

$$d_{\mathbb{B}_c}(\boldsymbol{z}_1, \boldsymbol{z}_2) = \frac{2}{\sqrt{c}} \tanh^{-1}\left(\sqrt{c}\left\|\frac{-(1+2c\langle-\boldsymbol{z}_1, \boldsymbol{z}_2\rangle + c\|\boldsymbol{z}_2\|^2)\boldsymbol{z}_1 + (1-c\|\boldsymbol{z}_1\|^2)\boldsymbol{z}_2}{1 - 2c\langle\boldsymbol{z}_1, \boldsymbol{z}_2\rangle + c^2\|\boldsymbol{z}_1\|^2\|\boldsymbol{z}_2\|^2}\right\|\right). \quad (3)$$

For $c \to 0$, we can recover the properties of the Euclidean geometry since $\lim_{c\to 0} d_{\mathbb{B}_c}(\boldsymbol{z}_1, \boldsymbol{z}_2) = 2\|\boldsymbol{z}_1 - \boldsymbol{z}_2\|$. Since $\mathcal{T}_{\boldsymbol{z}}\mathbb{B}_c^d$ is isomorphic to $\mathbb{R}^d$, we can connect vectors in Euclidean space and hyperbolic space with the bijection between $\mathcal{T}_{\boldsymbol{z}}\mathbb{B}_c^d$ and $\mathbb{B}_c^d$ [88]. For $\boldsymbol{z} = \boldsymbol{0}$, the *exponential map* $\exp_{\boldsymbol{0}}^c : \mathcal{T}_{\boldsymbol{z}}\mathbb{B}_c^d \to \mathbb{B}_c^d$ and *logarithm map* $\log_{\boldsymbol{0}}^c : \mathbb{B}_c^d \to \mathcal{T}_{\boldsymbol{z}}\mathbb{B}_c^d$ have the closed form of

$$\exp_{\boldsymbol{0}}^c(\boldsymbol{v}) = \tanh\left(\sqrt{c}\|\boldsymbol{v}\|\right)\frac{\boldsymbol{v}}{\sqrt{c}\|\boldsymbol{v}\|}, \quad \log_{\boldsymbol{0}}^c(\boldsymbol{u}) = \frac{1}{\sqrt{c}}\tanh^{-1}\left(\sqrt{c}\|\boldsymbol{u}\|\right)\frac{\boldsymbol{u}}{\|\boldsymbol{u}\|}. \quad (4)$$

Alternatively, to guarantee the correctness of Poincaré distance computation, we can also process any Euclidean vector with a clipping module [66]

$$\text{clip}^c(\boldsymbol{v}) = \begin{cases} \boldsymbol{v}, & \text{if } \|\boldsymbol{v}\| < 1/\sqrt{c} \\ \left(\frac{1}{\sqrt{c}} - \epsilon\right)\frac{\boldsymbol{v}}{\|\boldsymbol{v}\|}, & \text{otherwise} \end{cases}, \quad (5)$$

so the clipped vector is within the Poincare disk. We set $\epsilon$ as a small positive number in practice.

**Definition 3.3** (Klein Model). Klein model $(\mathbb{K}_c^d, \mathfrak{g}_K)$ consists of an $1/\sqrt{c}$-radius open ball $\mathbb{K}_c^d = \{z \in \mathbb{R}^d : c\|z\|^2 < 1\}$ and a metric tensor $\mathfrak{g}_K$ different from $\mathfrak{g}_B$. Similar to the mean computation in Euclidean space, let $\gamma_i = 1/\sqrt{1 - c\|z_i\|^2}$, the Einstein midpoint of a group of Klein vectors $z_1, \ldots z_n \in \mathbb{K}_c^d$ has a simple expression of a weighted average

$$\text{HypAve}_K(z_1, \ldots z_n) = \sum_{i=1}^n \gamma_i z_i \bigg/ \sum_{i=1}^n \gamma_i. \tag{6}$$

We illustrate the relationship between the different hyperbolic models in Figure 3. The hyperboloid space models $d$-dimensional hyperbolic geometry on a $d + 1$-dimensional space. When $d = 2$, the Klein model is the tangent plane of the hyperboloid model at $(0, 0, 1)$, and the Poincaré disk shares the same support as the Klein disk, although shifted downwards and centered at the origin. Given a triangle on the hyperboloid model, its projection on the Klein model preserves the straight sides, but the projection of a line on the Poincaré model is a part of a circular arc or the diameter of the disk. Let $z_\mathbb{B}, z_\mathbb{K}$ be coordinates of $z$ under Poincaré and Klein model respectively, the prototype operations on $\mathbb{B}_c^d$ require transformations between $\mathbb{B}_c^d$ and $\mathbb{K}_c^d$ as

$$z_\mathbb{B} = \frac{z_\mathbb{K}}{1 + \sqrt{1 - c\|z_\mathbb{K}\|^2}}, \quad z_\mathbb{K} = \frac{2z_\mathbb{B}}{1 + c\|z_\mathbb{B}\|^2}. \tag{7}$$

For example, in Figure 3, $O'$ is the $\text{HypAve}_K$ of $A', B', C'$, and can be mapped back to the Poincaré disk to get the Poincaré prototype ($\text{HypAve}_B$) $O$ of points $A, B, C$ by a change of coordinates.

## 3.2  `HypStructure`: **Hyperbolic Structured Regularization**

At a high level, `HypStructure` uses a combination of two losses: a Hyperbolic Cophenetic Correlation Coefficient Loss (`HypCPCC`)), and a Hyperbolic centering loss (`HypCenter`) for embedding the hierarchy in the representation space. Below we describe the two components of `HypStructure`. The pseudocode of `HypStructure` is shown in Algorithm 1 in Appendix B.2.

**HypCPCC (Hyperbolic Cophenetic Correlation Coefficient):** We extend the $\ell_2$-CPCC methodology in [104] to the hyperbolic space in `HypCPCC`. Three major steps of `HypCPCC` are (i) map Euclidean vectors to Poincaré space (ii) compute class prototypes (iii) use Poincaré distance for CPCC. Specifically, we first project each $z_i \in \mathbb{R}^d$ to $\mathbb{B}_c^d$, and compute the Poincaré centroid for each vertex of $\mathcal{T}$ using hyperbolic averaging as shown in Equation (6) and Equation (7). Alternatively, we can also compute Euclidean centroids $\overline{z} = \frac{1}{|\mathcal{D}_i|} \sum_{z \in \mathcal{D}_i} z$ for each vertex, and project each $\overline{z} \in \mathbb{R}^d$ to $\mathbb{B}_c^d$ either by $\exp_0^c$ or $\text{clip}^c$. After the computation of hyperbolic centroids, we use the pairwise distances between all vertex pairs in $\mathcal{T}$ in the Poincaré ball, to compute the `HypCPCC` loss using Equation (1) by setting $\rho = d_{\mathbb{B}_c}$.

**HypCenter (Hyperbolic Centering):** Inspired by Sarkar's low-distortion construction [75] that places the root node of a tree at the origin, we propose a centering loss for this positioning, that enforces the representation of the root node to be close to the center of the Poincaré disk, and the representations of its children to be closer to the border of Poincaré disk. We enforce this constraint by minimizing the norm of the hyperbolic representation of the root node as $\ell_{\text{center}} = \|\text{HypAve}_B(\exp_0^c(z_1), \ldots, \exp_0^c(z_N))\|$. Alternatively, for centroids computed in the Euclidean space and mapped to the Poincaré disk, we minimize $\ell_{\text{center}} = \left\| 1/N \sum_{i=1}^N f_\theta(x_i) \right\|$ directly due to the monotonicity of $\tanh(\cdot)$ in the exponential map.

Finally, for $\alpha, \beta > 0$, we can learn the *hierarchy-informed* representations by minimizing

$$\mathcal{L}(\mathcal{D}) = \sum_{(x,y) \in \mathcal{D}} \ell_{\text{Flat}}(x, y, \theta) - \alpha \cdot \text{HypCPCC}(d_\mathcal{T}, d_{\mathbb{B}_c}) + \beta \cdot \ell_{\text{center}}(x, \theta), \tag{8}$$

where $\ell_{\text{Flat}}$ is a standard classification loss, such as the `CE` loss or the `SupCon` loss.

**Time Complexity:** In a batch computation setting with a batch size $b$ and the number of classes (leaf nodes) as $k$, the computational complexity for a `HypStructure` computation to embed the full tree will still be $O(d \cdot \min\{b^2, k^2\})$, which is the same as the complexity of a Euclidean *leaf-only* CPCC. The additional knowledge gained from internal nodes allows us to reason about the relationship between higher-level concepts, and the hyperbolic representations help in achieving a low distortion of hierarchical information for better performance in downstream tasks.

# 4 Experiments

We conduct extensive experiments on several large-scale image benchmark datasets to evaluate the performance of HypStructure as compared to the Flat and $\ell_2$-CPCC baselines for hierarchy embedding, classification, and OOD detection tasks.

**Datasets and Setup**   Following the common benchmarks in the literature, we consider three real-world vision datasets, namely CIFAR10, CIFAR100 [45] and ImageNet100 [59] for training, which vary in scale, number of classes, and number of images per class. We construct the ImageNet100 dataset by sampling 100 classes from the ImageNet-1k [72] dataset following [59]. For CIFAR100, a three-level hierarchy is available with the dataset release [45]. Since no hierarchy is available for the CIFAR10 and ImageNet100 datasets, we construct a hierarchy for CIFAR10 manually in Figure 2. For ImageNet100, we create a subtree from the WordNet [19] given 100 classes as leaves. More details regarding the datasets, network, training and setup are provided in the Appendix B.4.

## 4.1 Quality of Hierarchical Information

Table 1: Evaluation of hierarchical information distortion and classification accuracy using SupCon [39] as $\ell_{\text{Flat}}$. All metrics are reported as mean (standard deviation) over 3 seeds.

| Dataset (Backbone) | Method | Distortion of Hierarchy | | Classification Accuracy | |
|---|---|---|---|---|---|
| | | $\delta_{rel}$ ($\downarrow$) | CPCC ($\uparrow$) | Fine ($\uparrow$) | Coarse ($\uparrow$) |
| CIFAR10 (ResNet-18) | Flat | 0.232 (0.001) | 0.573 (0.002) | 94.64 (0.12) | 99.16 (0.04) |
| | $\ell_2$-CPCC | 0.174 (0.002) | 0.966 (0.001) | 94.47 (0.13) | 98.91 (0.02) |
| | HypStructure | **0.094 (0.003)** | **0.992 (0.001)** | **94.79 (0.14)** | **99.18 (0.04)** |
| CIFAR100 (ResNet-34) | Flat | 0.209 (0.002) | 0.534 (0.119) | 74.96 (0.14) | 84.15 (0.19) |
| | $\ell_2$-CPCC | 0.213 (0.006) | **0.779 (0.002)** | 76.07 (0.19) | 85.28 (0.32) |
| | HypStructure | **0.127 (0.016)** | 0.766 (0.007) | **76.68 (0.22)** | **86.01 (0.13)** |
| ImageNet100 (ResNet-34) | Flat | 0.168 (0.003) | 0.429 (0.002) | 90.01 (0.07) | 90.77 (0.11) |
| | $\ell_2$-CPCC | 0.213 (0.009) | 0.834 (0.002) | 89.57 (0.38) | 90.34 (0.28) |
| | HypStructure | **0.134 (0.001)** | **0.841 (0.001)** | **90.12 (0.01)** | **90.84 (0.02)** |

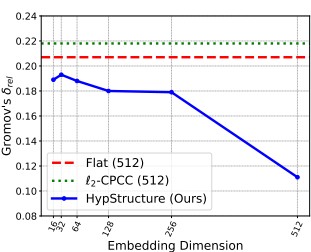

Figure 4: Evaluation of distortion vs feature dimensions for HypStructure.

First, to assess the *tree-likeness* of the learnt representations, we measure the Gromov's hyperbolicity $\delta_{rel}$ [23, 1, 38, 40] of the features in Table 1. Lower $\delta_{rel}$ indicates higher tree-likeness and a perfect tree metric space has $\delta_{rel} = 0$ (more details in Appendix B.5). To also evaluate the correspondence of the feature distances with ground truth tree metrics, we compute CPCC on test sets. We observe that HypStructure reduces distortion of hierarchical information over Flat by upto **59.4%** and over $\ell_2$-CPCC by upto **45.4%**, while also consistently improving the test CPCC for most datasets.

We also perform a qualitative analysis of the learnt representations from HypStructure on the CIFAR10 dataset, and visualize them in a Poincaré disk using UMAP [57] in Figure 5a. We can observe clearly that the samples for fine classes arrange themselves in the Poincaré disk based on the hierarchy tree as seen in Figure 2, being closer to the classes which share a *coarse* class parent.

To examine the impact of feature dimension on the representative capacity of the hyperbolic space, we vary the feature dimension for HypStructure and compute the $\delta_{rel}$ for each learnt feature. Comparing the distortion of features with the Flat and $\ell_2$-CPCC settings in Figure 4, we observe that $\delta_{rel}$ decreases consistently with increasing dimensions, implying that high dimension features using HypStructure are more tree-like, and better than Flat and $\ell_2$-CPCCs' 512-dimension baselines.

## 4.2 Classification

Following [104], we treat leaf nodes in the hierarchy as *fine* classes and their parent nodes as *coarse* classes. To evaluate the quality of the learnt representations, we perform a classification task on the fine and coarse classes using a kNN-classifier following [27, 95, 5, 110] and report the performance on the three datasets in Table 1. We observe that HypStructure leads to upto **2.2%** improvements over Flat and upto **0.8%** improvements over $\ell_2$-CPCC on both fine and coarse accuracy. We also visualize the learnt test features from Flat vs HypStructure on the CIFAR100 dataset using Euclidean t-SNE [89] and show the visualizations in Figure 5b and Figure 5c respectively. We observe that HypStructure leads to sharper and more discriminative representations in Euclidean space. Additionally, we see that the fine classes belonging to a coarse class (the same shades of colors)

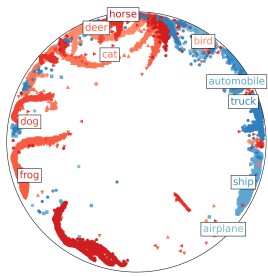

(a) Hyperbolic UMAP:
`HypStructure`

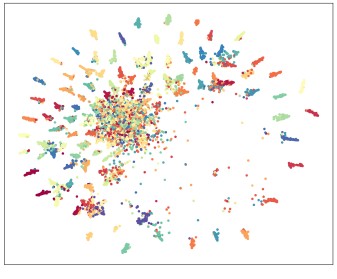

(b) Euclidean t-SNE: Flat

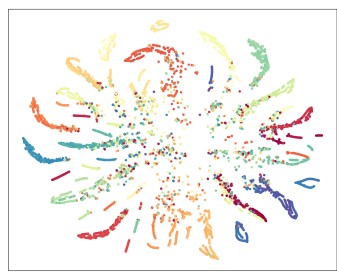

(c) Euclidean t-SNE:
`HypStructure`

Figure 6: Left: Hyperbolic UMAP visualization of CIFAR10's `HypStructure` representation on Poincaré disk. Middle and Right: t-SNE visualization of learnt representations on CIFAR100.

which are semantically closer in the label hierarchy, are grouped closer and more compactly in the feature space as well, as compared to Flat. We also perform evaluations using the linear evaluation protocol [39] and observe an identical trend in the accuracy, we report these results in Appendix C.1.

### 4.3 OOD Detection

In addition to leveraging the hierarchy explicitly for the purpose of learning tree-like ID representations, we argue that a structured separation of features in the hyperbolic space as enforced by `HypStructure` is helpful for the OOD detection task as well. To verify our claim, we perform an exhaustive evaluation on 9 real-world OOD datasets and demonstrate that `HypStructure` leads to improvements in the OOD detection AUROC. We share more details below.

| Method | OOD Dataset AUROC (↑) | | | | | Overall AUROC | |
|---|---|---|---|---|---|---|---|
| | SVHN | Textures | Places365 | LSUN | iSUN | Avg.(↑) | B.C.(↑) |
| ProxyAnchor [41] | 82.43 | 84.99 | 79.84 | 91.68 | 84.96 | 84.78 | 51.42 |
| CE + SimCLR [94] | 94.45 | 82.01 | 71.48 | 89.00 | 83.82 | 84.15 | 31.42 |
| CSI [85] | 92.65 | 86.47 | 76.27 | 83.78 | **84.98** | 84.83 | 40.00 |
| CIDER [61] | 95.16 | 90.42 | 73.43 | 96.33 | 82.98 | 87.67 | 60.00 |
| SSD+ (SupCon) [76] | 94.19 | 86.18 | **79.90** | 85.18 | 84.08 | 85.90 | 54.28 |
| KNN+ (SupCon) [83] | 92.78 | 88.35 | 77.58 | 89.30 | 82.69 | 86.14 | 40.00 |
| $\ell_2$-CPCC [104] | 93.08 | **90.45** | 77.21 | 82.77 | 82.79 | 85.26 | 40.00 |
| `HypStructure` | **95.97** | 88.43 | 78.12 | **97.01** | 84.51 | **88.81** | **82.85** |

(a) OOD detection performance with CIFAR100 as ID dataset.

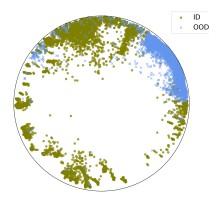

(b) CIFAR100 (ID) vs. SVHN (OOD).

Figure 7: Left: OOD detection score across various datasets on the CIFAR100 ID dataset. Right: Hyperbolic UMAP of the CIFAR100 (ID) test vs SVHN (OOD) test features learnt from `HypStructure` with a clear separation in the Poincaré disk.

#### 4.3.1 Problem Setting

Out-of-distribution (OOD) data refers to samples that do not belong to the in-distribution (ID) and whose label set is disjoint from $\mathcal{Y}^{\text{in}}$ and therefore should not be predicted by the model. Therefore the goal of the OOD detection task is to design a methodology that can solve a binary problem of whether an incoming sample $x \in \mathcal{X}$ is from $P_{\mathcal{X}}$ i.e. $y \in \mathcal{Y}^{\text{in}}$ (ID) or $y \notin \mathcal{Y}^{\text{in}}$ (OOD).

**OOD datasets** We evaluate on 5 OOD image datasets when CIFAR10 and CIFAR100 are used as the ID datasets, namely SVHN [65], `Places365` [109], `Textures` [9], `LSUN` [102], and `iSUN` [99], and 4 large scale OOD test datasets, specifically SUN [102], `Places365` [109], `Textures` [9] and `iNaturalist` [90] when ImageNet100 is used as the ID dataset. This subset of datasets is prepared by [59] and is created with overlapping classes from ImageNet-1k removed from these datasets to ensure there is no overlap in the distributions.

**OOD detection scores** While several scores have been proposed for the task of OOD detection, we evaluate our proposed method using the Mahalanobis score [76], computed by estimating the mean

Table 2: OOD detection AUROC with CIFAR10 and ImageNet100 as ID.

| Method | AUROC | Method | AUROC |
|---|---|---|---|
| **CIFAR10** | | **ImageNet100** | |
| SSD+ | 97.38 | SSD+ | 92.46 |
| KNN+ | 97.22 | KNN+ | 92.74 |
| $\ell_2$-CPCC | 76.67 | $\ell_2$-CPCC | 91.33 |
| HypStructure | **97.75** | HypStructure | **93.83** |

and covariance of the in-distribution training features. The Mahalanobis score is defined as

$$s(\boldsymbol{x}) = (f(\boldsymbol{x}) - \mu)^\top \Sigma^{-1} (f(\boldsymbol{x}) - \mu), \qquad (9)$$

where $\mu, \Sigma$ are the mean and covariance of in-distribution training features. [76] present the Mahalanobis score (eq. (9)) in a generalized version for multiple feature clusters. However, since they empirically observe that the single-cluster version achieves the highest performance [76], we will focus on this version. After computing the OOD detection scores, we measure the area under the receiver operating characteristic curve (AUROC) as the primary evaluation metric following [47, 76].

### 4.3.2 Main Results and Discussion

We report the AUROC averaged over all the OOD datasets (5 datasets for CIFAR10 and CIFAR100, 4 datasets for ImageNet100) in Figure 7a and Table 2 In addition to the Flat (SupCon) and $\ell_2$-CPCC baselines, we also compare our method with other state-of-the-art methods (see Appendix C.3.1 for more details about existing OOD detection methods). We observe that HypStructure leads to a consistent improvement in the OOD detection score, with upto **2%** in average AUROC. We also report the dataset-wise OOD detection results for the CIFAR100 ID dataset in Table 7a along with Average AUROC. To remove the bias in the Average AUROC metric towards any single dataset, we also evaluate the Borda Count (B.C.) [58] and report the same, along with a detailed comparison with more OOD detection methods in Table 7a, and Tables 6 and 7 in the Appendix C.3. We observe that HypStructure ranks in the highest performing methods consistently, thereby demonstrating a higher Borda Count as well. We additionally visualize the CIFAR100 (ID) vs SVHN (OOD) features learnt from HypStructure, using a hyperbolic UMAP visualization in Figure 7b. We observe that training with HypStructure leads to an improvement in the separation of ID vs OOD features in the Poincaré disk.

**Additional Experiments, Ablations and Visualizations:** More experiments using hyperbolic contrastive losses and hyperbolic networks, ablation studies on each component of HypStructure and additional visualizations can be found in Appendix C.

## 5 Eigenspectrum Analysis of Structured Representations

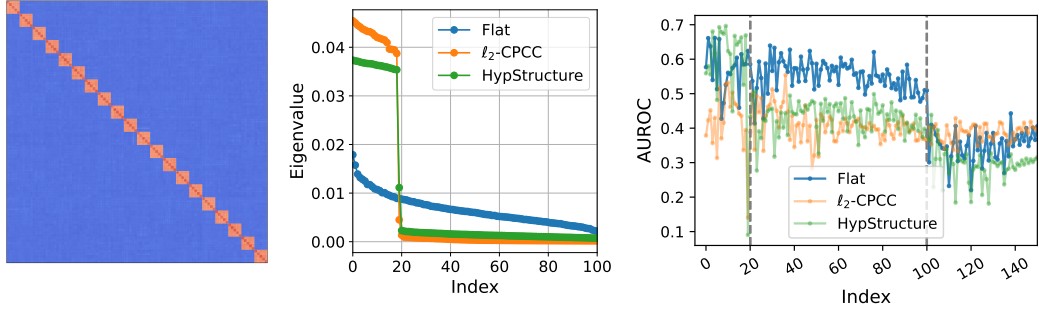

Figure 8: CIFAR100 as in-distribution dataset. Left (a): Hierarchical block pattern of $K$. Middle (b): Top 100 eigenvalues of $K$ for different representation. Right (c): OOD detection for CIFAR100 vs. SVHN with the top $k$-th principal component.

As seen in Figure 7a, we observe a significant improvement in the OOD detection performance using `HypStructure` with the Mahalanobis score eq. (9). After training a composite loss with CPCC till convergence, let us denote the matrix of the normalized in-distribution trained feature as $Z \in \mathbb{R}^{n \times d}$. Naturally, we inspect the eigenvalue properties of $\Sigma$ (i.e, $Z$), and observe that $K = ZZ^{\top} \in \mathbb{R}^{n \times n}$ exhibits a hierarchical block structure (Figure 8a) where the diagonal blocks have a significantly higher correlation than other off-diagonal values, leading us to the following theorem.

**Theorem 5.1** (**Eigenspectrum of Structured Representation with Balanced Label Tree**). *Let $\mathcal{T}$ be a balanced tree with height $H$, such that each level has $C_h$ nodes, $h \in [0, H]$. Let us denote each entry of $K$ as $r^h$ where $h$ is the height of the lowest common ancestor of the row and the column sample. If $r^h \geq 0, \forall h$, then: (i) For $h = 0$, we have $C_0 - C_1$ eigenvalues $\lambda_0 = 1 - r^1$. (ii) For $0 < h \leq H - 1$, we have $C_h - C_{h+1}$ eigenvalues $\lambda_h = \lambda_{h-1} + (r^h - r^{h+1})\frac{C_0}{C_h}$. (iii) The last eigenvalue is $\lambda_H = \lambda_{H-1} + C_0 r^H$.*

We defer the eigenspectrum analysis for an arbitrary label tree to Appendix A. Theorem 5.1 implies a **phase transition pattern** in the eigenspectrum. There always exists a significant gap in the eigenvalues representing each level of nodes in the hierarchy, and the eigenvalues corresponding to the coarsest level are the highest in magnitude. CIFAR100 has a balanced three-level label hierarchy where each coarse label has five fine labels as its children. In Figure 8b, we visualize the eigenspectrum of CIFAR100 for `HypStructure`, $\ell_2$-CPCC and the Flat objective. We observe a significant drop in the eigenvalues for features learnt from two hierarchical regularization approaches, $\ell_2$-CPCC and `HypStructure`, at approximately the 20th largest eigenvector (which corresponds to the number of coarse classes), whereas these phase transitions do not appear for standard flat features. We also observe that the magnitude of coarse eigenvalues are approximately at the same scale.

In summary, Theorem 5.1 helps us to formally characterize the difference between flat and structured representations. CPCC style (eq. (1)) regularization methods can also be treated as dimensionality reduction techniques, where the structured features can be explained mostly by the coarser level features. For the OOD detection setting, this property differentiates the ID and OOD samples at the coarse level itself using a lower number of dimensions, and makes the OOD detection task easier. We visualize the OOD detection AUROC on SVHN (OOD) corresponding to the CIFAR100 (ID) features with the top$-k$ principal component for different methods, in Figure 8c. We observe that for features learnt using `HypStructure`, accurately embedding the hierarchical information leads to the top 20 eigenvectors (corresponding to the coarse classes) being the most informative for OOD detection. Recall that CIDER [61] is a state-of-the-art method proposed specifically for improving OOD detection by increasing inter-class dispersion and intra-class compactness. We note that CPCC style (eq. (1)) methods can be seen as a generalization of CIDER on higher-level concepts, where the same rules are applied for coarse labels as well, along with the fine classes. When the ID and OOD distributions are far enough, using coarse level feature might be sufficient for OOD detection.

# 6   Related Work

**Learning with Label Hierarchy.**   Several recent works have explored how to leverage hierarchical information between classes for various purposes such as relational consistency [14], designing specific hierarchical classification architectures [101, 25, 68], hierarchical conditioning of the logits [13], learning order preserving embeddings [15], and improving classification accuracy [91, 86, 48, 49, 108, 34]. The proposed structural regularization framework in [104] offers an interesting approach to embed a tree metric to learn structured representations through an *explicit* objective term, although they rely on the $\ell_2$ distance, which is less than ideal for learning hierarchies.

**Hyperbolic Geometry.**   [66] first proposed using the hyperbolic space to learn hierarchical representations of symbolic data such as text and graphs by embedding them into a Poincaré ball. Since then, the use of hyperbolic geometry has been explored in several different applications. [40] proposed a hyperbolic image embedding for few-shot learning and person re-identification. [20] proposed hyperbolic neural network layers, enabling the development of hybrid architectures such as hyperbolic convolutional neural networks [78], graph convolutional networks [12], hyperbolic variational autoencoders [56] and hyperbolic attention networks [24]. Additionally, these hybrid architectures have also been explored for different tasks such as deep metric learning [18, 100], object detection [46] and natural language processing [16]. There have also been several investigations into the properties of hyperbolic spaces and models such as low distortion [75], small generalization error [84] and

representation capacity [62]. However, none of these works have leveraged hyperbolic geometry for *explicitly* embedding a hierarchy in the representation space via structured regularization, and usually attempt to leverage the underlying hierarchy *implicitly* using hyperbolic models.

# 7 Discussion and Future Work

In this work, we introduce `HypStructure`, a simple-yet-effective structural regularization framework for incorporating the label hierarchy into the representation space using hyperbolic geometry. In particular, we demonstrate that accurately embedding the hierarchical relationships leads to empirical improvements in both classification as well as the OOD detection tasks, while also learning *hierarchy-informed* features that are more interpretable and exhibit less distortion with respect to the label hierarchy. We are also the first to formally characterize properties of *hierarchy-informed* features via an eigenvalue analysis, and also relate it to the OOD detection task, to the best of our knowledge. We acknowledge that our proposed method depends on the availability or construction of an external hierarchy for computing the HypCPCC objective. If the hierarchy is unavailable or contains noise, this could present challenges. Therefore, it is important to evaluate how injecting noisy hierarchies into CPCC-based methods impacts downstream tasks. While the current work uses the Poincaré ball model, exploring the representation trade-offs and empirical performances using other models of hyperbolic geometry in `HypStructure`, such as the Lorentz model [67] is an interesting future direction. Further theoretical investigation into establishing the error bounds of CPCC style structured regularization objectives is of interest as well.

## Acknowledgement

SZ and HZ are partially supported by an NSF IIS grant No. 2416897. HZ would like to thank the support from a Google Research Scholar Award. MY was supported by MEXT KAKENHI Grant Number 24K03004. We would also like to thank the reviewers for their constructive feedback during the review process. The views and conclusions expressed in this paper are solely those of the authors and do not necessarily reflect the official policies or positions of the supporting companies and government agencies.

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

## Appendix

This appendix is segmented into the following key parts.

1. **Section A** continues the analysis of the eigenspectrum of CPCC-optimized representation matrix and generalizes it for an arbitrary label tree.

2. **Section B** discusses additional details about our proposed method `HypStructure`, its implementation and broader impact of our work. In particular, an overview of the method is first provided, and then we describe hyperparameter settings of our method and the main baselines, followed by extra dataset details and explanation of evaluation metrics.

3. **Section C** reports ablation studies, detailed results on OOD detection and provides additional experimental results and visualizations not included in the main paper due to lack of space.

## A    Details of Eigenspectrum Analysis

In this section, we first introduce some notations, discuss the setup for our analysis, followed by preliminary lemmas, and then characterize the eigenspectrum of CPCC-based structured representations in Theorem A.2 for an arbitrary label tree and Theorem 5.1 for a balanced tree presented in the main body.

**Proof Sketch**    The proof of Theorem A.2 and Theorem 5.1 relies on the important observation of a *hierarchical* block structure of the covariance matrix of CPCC-regularized features, as shown in Figure 8a, which will also be supported by Lemma A.1 and Corollary A.1. Theorem A.1 [3] and Lemma A.2 characterize the eigenvalues of a block correlation matrix induced from a basic tree where the matrix only has three types of values: diagonal values of 1s, one for within group entry, and another for across group entry. Larger within group entries lead to the larger eigenvalues. Theorem A.1 [3] and Lemma A.2 are then used as the base case for the induction proof of Theorem 5.1. For an arbitrary tree, in Theorem A.2, we use Weyl's Theorem [93] to bound the gap between within group entries and across group entries that leads to the phase transition of eigenvalues.

**Setup details**    After training with the `HypStructure` loss till convergence, let us denote the feature matrix as $Z \in \mathbb{R}^{n \times d}$, where each row of $Z$ is a $d$-dimensional vector of an in distribution training sample, and the CPCC is maximized to 1. We let $C_0 = n, C_1, C_2, \ldots, C_H = 1$ be the number of class labels at height $h$ of the tree $\mathcal{T}$. Following the standard pre-processing steps in OOD detection [76], we assume that the features are standardized and normalized so that $\mathbb{E}[Z] = \mathbf{0}$ and $\|Z_i\|_2 = 1, \forall i$. Besides, we assume that in $\mathcal{T}$, the distance from root node to each leaf node is the same. Otherwise, following [74], we can insert dummy parents or children into the tree to make sure vertices at the same level have similar visual granularity. We then apply CPCC to each node in the extended tree, where each leaf node is one sample. We note that although this is slightly different from the implementation where the leaf nodes are fine class nodes, the distance for samples within fine classes are automatically minimized by classification loss like cross-entropy and supervised contrastive loss.

Given these assumptions, we want to analyze the eigenspectrum of the inverse sample covariance matrix $\frac{1}{n-1}Z^\top Z$, which is the same as investigating the eigenvalues of $K = ZZ^\top$ where $Z$ is ordered by classes at all levels, i.e., samples having the same fine-grained labels and coarse labels should be placed together. This is because the matrix scaling and permutation will not change the order of singular values.

Since CPCC (eq. (1)) is a correlation coefficient, when it is maximized, the $n$ by $n$ pairwise Poincaré distance matrix is perfectly correlated with the ground truth pairwise *tree-metric* matrix, where each entry is the tree distance between two samples on the tree, no matter we apply CPCC to leaves or all vertices. This implies that in the similarity matrix $K$, the relative order of entries are the opposite of tree matrix, and it is trivial to show it as follows

**Lemma A.1.** The relative order of entries in $K$ will be the reverse of the order in tree distance.

*Proof.* When $\|u\| = \|v\| = 1$, $\ell_2(u,v)^2 = \|u-v\|_2^2 = \|u\|^2 + \|v\|^2 - 2\langle u,v\rangle = 2 - 2\langle u,v\rangle$. Now considering the CPCC computation, if the CPCC is maximized, the pairwise Euclidean matrix is of

the scalar factor of the tree distance matrix. Since each entry of $K$ is the dot product of two samples, the relative order in $K$ is the opposite. ∎

**Corollary A.1.** If we use the Poincaré distance (eq. (3) in CPCC and let the curvature constant $c = 1$, the statement of cosine distance in Lemma A.1 still holds.

*Proof.* Since the Poincaré distance (eq. (3)) is only defined for vectors with magnitude less than 1, let us consider the case where before the clipping operation, both $u$ and $v$ are outside the unit ball. After applying clip[1], $\|u\| = \|v\| = 1 - \epsilon$, where $\epsilon$ is a small constant ($10^{-5}$). Then $\|u\|^2 = (1 - \epsilon)^2 = 1 - 2\epsilon + \epsilon^2$. Define $2\epsilon - \epsilon^2$ as $\xi$, making $\|u\|^2 := 1 - \xi$ where $\xi$ is also a small constant such that $O(\xi^2)$ is negligible.

$$\text{Poincaré}(u, v) = 2 \ln \frac{\|u - v\| + \sqrt{\|u\|^2 \|v\|^2 - 2u \cdot v + 1}}{\sqrt{(1 - \|u\|^2)(1 - \|v\|^2)}}$$

$$= 2 \ln \frac{\|u - v\| + \sqrt{2 - 2u \cdot v - 2\xi + \xi^2}}{\xi}$$

$$\approx 2 \ln \frac{\|u - v\| + \sqrt{2 - 2u \cdot v - 2\xi}}{\xi}$$

$$= 2 \ln \frac{\|u - v\| + \sqrt{\|u\|^2 + \|v\|^2 - 2u \cdot v}}{\xi}$$

$$= 2 \ln \frac{2 \|u - v\|}{\xi}$$

We can see that the Poincaré distance monotonically increases with Euclidean distances $\|u - v\|$. This property ensures the relative order of any two entries for Euclidean CPCC and Poincare CPCC matrices in $K$ to be the same. Then, we can argue about the structure of $K$, either Euclidean or Poincare, to have the hierarchical diagonalized structure as in Figure 8a. So any statement applied for a Poincaré version of CPCC will also hold for the Euclidean CPCC counterpart. ∎

For each level of the tree, due to the optimization of CPCC loss, the corresponding off diagonal entries of $K$, which represent the intra-level-class similarities, are much smaller than inter-level-class values. We thus have a symmetric similarity matrix that takes on the following structure, where the red regions are greater than orange regions, which are further greater than the blue regions.

$$K = \begin{bmatrix} 1 & r_{11}^1 & r_{12}^2 & r_{12}^2 & r_{13}^3 & r_{13}^3 & r_{14}^3 & r_{14}^3 & \cdots \\ r_{11}^1 & 1 & r_{12}^2 & r_{12}^2 & r_{13}^3 & r_{13}^3 & r_{14}^3 & r_{14}^3 & \cdots \\ r_{12}^2 & r_{12}^2 & 1 & r_{22}^1 & r_{23}^3 & r_{23}^3 & r_{24}^3 & r_{24}^3 & \cdots \\ r_{12}^2 & r_{12}^2 & r_{22}^1 & 1 & r_{23}^3 & r_{23}^3 & r_{24}^3 & r_{24}^3 & \cdots \\ r_{13}^3 & r_{13}^3 & r_{23}^3 & r_{23}^3 & 1 & r_{33}^1 & r_{34}^2 & r_{34}^2 & \cdots \\ r_{13}^3 & r_{13}^3 & r_{23}^3 & r_{23}^3 & r_{33}^1 & 1 & r_{34}^2 & r_{34}^2 & \cdots \\ r_{14}^3 & r_{14}^3 & r_{24}^3 & r_{24}^3 & r_{34}^2 & r_{34}^2 & 1 & r_{44}^1 & \cdots \\ r_{14}^3 & r_{14}^3 & r_{24}^3 & r_{24}^3 & r_{34}^2 & r_{34}^2 & r_{44}^1 & 1 & \cdots \\ \cdots & \cdots & \cdots & \cdots & \cdots & \cdots & \cdots & \cdots & \cdots \end{bmatrix}$$

Each non-diagonal entry is called $r_{ij}^h$ where $i, j$ are the index of the diagonal block, or the *finest* label id of one sample, and $h$ is the height of the lowest common ancestor of the two samples in the row and the column. Since every two leaves sharing the lowest common ancestor of the same height have the same tree distance, each entry of $K$ with the same superscript will be the same so we can drop the $i, j$ subscript in the notation. The size of each block is defined by the number of samples within one label. Then, the shown submatrix of $K$ corresponds to the following tree in Figure 9. Next, we present several useful lemmas and theorems.

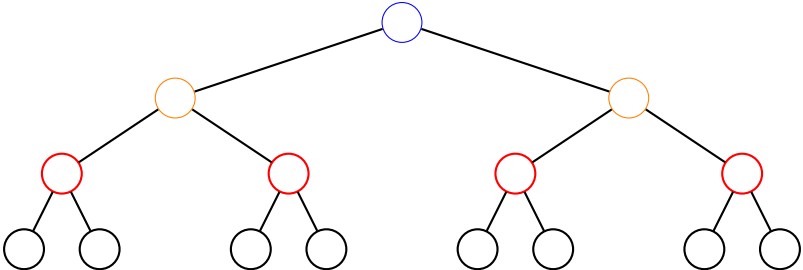

Figure 9: Subtree corresponds to the shown submatrix of $K$.

**Theorem A.1** ([3]). Let $R$ be a $p \times p$ full-rank correlation matrix that has a $k$-group block structure, with groups of size $p_i$ ($i = 1 : k$, $\sum_{i=1}^{k} p_i = p$). Let $r_{ii}$ be the correlation for any pair of different variables within group $i$. Let $r_{ij}$ be the common correlation between any variable in group $i$ and $j$. Denote the mean of the $i$-th diagonal block of $R$ by $\overline{R_i} = (1/p_i)(1 + (p_i - 1)r_{ii})$. Then:

1. $R$ has $p_i - 1$ eigenvalues $1 - r_{ii}$ ($i = 1 : k$).

2. The rest of the eigenvalues are those from $k \times k$ symmetric matrix $A$ whose diagonal elements are $a_{ii} = p_i \overline{R_i}$ and whose off-diagonal elements are $a_{ij} = \sqrt{p_i \cdot p_j} r_{ij}$.

**Lemma A.2.** Given $d$ by $d$ matrix $M$ where $M_{ii} = 1, \forall i \in [d]$, and $M_{ij} = p$ otherwise, i.e.,

$$M = \begin{bmatrix} 1 & p & \cdots & \cdots & p \\ p & \ddots & & & \vdots \\ \vdots & & 1 & & \vdots \\ \vdots & & & \ddots & p \\ p & \cdots & \cdots & p & 1 \end{bmatrix}$$

it has eigenvalues $\lambda_1 = 1 + p(d - 1)$ and $\lambda_2 = \cdots = \lambda_d = 1 - p$.

*Proof.* Using the definition of eigenvalues, we want to compute the determinant of matrix

$$\begin{bmatrix} 1 - \lambda & p & \cdots & \cdots & p \\ p & \ddots & & & \vdots \\ \vdots & & 1 - \lambda & & \vdots \\ \vdots & & & \ddots & p \\ p & \cdots & \cdots & p & 1 - \lambda \end{bmatrix}$$

Adding the second till the last row to the first row, we have

$$\begin{bmatrix} 1 - \lambda + (d-1)p & 1 - \lambda + (d-1)p & \cdots & \cdots & 1 - \lambda + (d-1)p \\ p & & \ddots & & \vdots \\ \vdots & & 1 - \lambda & & \vdots \\ \vdots & & & \ddots & p \\ p & & \cdots & \cdots & p & 1 - \lambda \end{bmatrix}$$

Dividing the first row by $1 - \lambda + (d - 1)p$, we now have

$$\begin{bmatrix} 1 & 1 & \cdots & \cdots & 1 \\ p & \ddots & & & \vdots \\ \vdots & & 1 - \lambda & & \vdots \\ \vdots & & & \ddots & p \\ p & \cdots & \cdots & p & 1 - \lambda \end{bmatrix}$$

Subtracting the second till the last row by $p$ times the first row results in an upper triangular matrix

$$
\begin{bmatrix}
1 & 1 & \cdots & \cdots & 1 \\
0 & 1-\lambda-p & & & 0 \\
\vdots & & 1-\lambda-p & & \vdots \\
\vdots & & & \ddots & 0 \\
0 & \cdots & \cdots & 0 & 1-\lambda-p
\end{bmatrix}
$$

Thus, $\det(M-\lambda I) = (1-\lambda+(d-1)p)(1-\lambda-p)^{d-1}$. $\blacksquare$

Notice that Lemma A.2 is a special case of Theorem A.1 where the label tree is a two level basic tree with the root node and $d$ leaves in the second label all being the direct children of the root node. Now we can leverage Theorem A.1 and Lemma A.2 to investigate the eigenspectrum of $K$ by proving the following theorem:

**Theorem A.2 (Eigenspectrum of CPCC-based Structured Representation).** If $\mathcal{T}$ is a tree whose root node has height $H$ where each level has $C_h$ nodes, $h \in [0, H]$. $K = ZZ^\top$ is a block structured correlation matrix as a result of CPCC optimization, where each off-diagonal entry $\in [-1, 1]$ can be written as $r^h$ and $h$ is the height of the lowest common ancestor of the $i$-th row and the $00j$-th column sample. Let $\Delta = r^1 - r^h$, $p_i, i \in [C_h]$ be the number of children for nodes at height $h$, and $p_{\max}$ be the maximum. For any $h \geq 1$, if $r^h \geq M \geq 0$, $r^{h+1} \leq m$, then

(i) We have $C_0 - C_h$ eigenvalues, that come from the eigenvalues of a $C_0 \times C_0$ matrix sharing the same $C_h$ of $p_i \times p_i$ diagonal blocks from $K$ subtracting $r^h$, and off diagonal values are all zero.

(ii) The rest of $C_h$ eigenvalues come from a $C_h \times C_h$ matrix, whose diagonal entries are the mean of each $p_i \times p_i$ diagonal block from $K$, and the off diagonal entries are $\sqrt{p_i p_j}\, r_{ij}$ where $r_{ij}$ is the correlation between node $i$ and node $j$ of height $h$.

(iii) If $m \leq \frac{M-2\Delta(p_{\max}-1)}{p_{\max}(C_h-1)}$, $C_0 - C_h$ eigenvalues are all smaller than $C_h$ eigenvalues.

*Proof.* Part (i) and (ii) can be extended from the proof of Theorem A.1. Let $G$ be the $n \times C_h$ matrix where $G_{ij} = 1$ if the $i$-th sample is in group $j$, otherwise $G_{ij} = 0$. For any $n - C_h$ eigenvectors in the orthogonal complement of the column space of $G$, the eigenvector of $K$ is also the eigenvector of

$$
\begin{bmatrix}
K_1 & 0 & \cdots & 0 \\
0 & K_2 & \cdots & 0 \\
\vdots & \vdots & \ddots & \vdots \\
0 & 0 & \cdots & K_k
\end{bmatrix}
$$

where due to the hierarchical structure of block matrix, $K_i$ has the format of

$$
\begin{bmatrix}
1-r^h & r^1-r^h & \cdots & r^j-r^h & 0 & \cdots & 0 \\
r^1-r^h & 1-r^h & \cdots & 0 & 0 & \cdots & 0 \\
\vdots & \vdots & \ddots & \vdots & \vdots & & \vdots \\
r^j-r^h & 0 & \cdots & 1-r^h & 0 & \cdots & 0 \\
0 & 0 & \cdots & 0 & 1-r^h & \cdots & 0 \\
\vdots & \vdots & & \vdots & \vdots & \ddots & \vdots \\
0 & 0 & \cdots & 0 & 0 & \cdots & 1-r^h
\end{bmatrix}
$$

The rest of $C_h$ eigenvectors come from the symmetric $C_h \times C_h$ matrix $A = (G^\top G)^{-1/2}(G^\top K G)(G^\top G)^{-1/2}$, by some basica algebra, we know $a_{ij} = \frac{1}{\sqrt{p_i}} \cdot$ (sum of all $r_{ij}$ entries in $p_i \times p_j$ block) $\cdot \frac{1}{\sqrt{p_j}}$. For more details, we refer the reader to Theorem 3.1 in [3] where $C_1 = k$ using their notation.

Since the largest absolute value of $K$'s eigenvalues, is bounded above by the largest row sum of the absolute values of $K$ [35], first $n - C_h$ eigenvectors are bounded above by $U = \max_i(1 - r^h) + (p_i - 1)(r^1 - r^h) = (1 - r^h) + (p_{\max} - 1)\Delta$.

On the other hand, for the rest of $C_h$ eigenvalues, we analyze matrix $A$:

$$A \leq A_1 + A_2$$

$$:= \begin{bmatrix} 1 + (p_1 - 1)r^h & 0 & \cdots & 0 \\ 0 & 1 + (p_2 - 1)r^h & \cdots & 0 \\ \vdots & \vdots & \ddots & \vdots \\ 0 & 0 & \cdots & 1 + (p_k - 1)r^h \end{bmatrix}$$

$$+ \begin{bmatrix} (p_{\max} - 1)(r^1 - r^h) & p_{\max}m & \cdots & p_{\max}m \\ p_{\max}m & (p_{\max} - 1)(r^1 - r^h) & \cdots & p_{\max}m \\ \vdots & \vdots & \ddots & \vdots \\ p_{\max}m & p_{\max}m & \cdots & (p_{\max} - 1)(r^1 - r^h) \end{bmatrix}$$

The inequality comes from the effect of the maximization of CPCC that $r^1 \geq \cdots r^h \geq r^{h+1} \geq \cdots r^H$ and $r^h \leq m$. The eigenvalues of $A_1, A_2$ have the analytical form, where $A_1$'s eigenvalues have the form of $1 + (p_i - 1)r^h$ and $A_2$'s eigenvalues can be derived by Lemma A.2. By Weyl's inequality [93], the minimum of these $C_h$ eigenvalues is at least $L = (1 + (p_{\min} - 1)r^h) - [(p_{\max} - 1)\Delta - p_{\max}m + kp_{\max}m] \geq (1 + (1 - 1)r^h) - [(p_{\max} - 1)\Delta - p_{\max}m + kp_{\max}m]$.

To guarantee eigenvalues from Part (ii) are larger, we want $L \geq U$. We solve this inequality with $m$, and we will get the desired range of $m$. ∎

When $r^h = r^1$ in Theorem A.2, we have $\Delta = 0$. Therefore, for a three level basic tree with only $r^1, r^2$, if $m \leq M/(p_{\max}(C_1 - 1))$, $C_0 - C_1$ eigenvalues are all smaller than $C_1$ eigenvalues. In general, we have shown that when $m$, i.e., the across group similarity is sufficiently small, the eigenvalue gap always exists. When the label tree $\mathcal{T}$ is balanced, we can further specify the expression of each eigenvalue and the amount of eigenvalue gaps.

We now formally restate the Theorem 4.1 from the main paper and give its proof.

**Theorem 5.1 (Eigenspectrum of Structured Representation with Balanced Label Tree).** *Let $\mathcal{T}$ be a balanced tree with height $H$, such that each level has $C_h$ nodes, $h \in [0, H]$. Let us denote each entry of $K$ as $r^h$ where $h$ is the height of the lowest common ancestor of the row and the column sample. If $r^h \geq 0, \forall h$, then: (i) For $h = 0$, we have $C_0 - C_1$ eigenvalues $\lambda_0 = 1 - r^1$. (ii) For $0 < h \leq H - 1$, we have $C_h - C_{h+1}$ eigenvalues $\lambda_h = \lambda_{h-1} + (r^h - r^{h+1})\frac{C_0}{C_h}$. (iii) The last eigenvalue is $\lambda_H = \lambda_{H-1} + C_0 r^H$.*

*Proof.* From Corollary A.1, we know that $K \in \mathbb{R}^{C_0 \times C_0}$ has a block-wise structure.

Since all statements are presented recursively, we prove the theorem by structural induction on the height of the tree.

The base case is Lemma A.2 with a two level hierarchy tree where only (i) and (iii) are applicable, and $p = r^1, C_0 = d, C_1 = 1$. By Lemma A.2, $K$ has $C_0 - 1$ eigenvalues as $\lambda_0 = 1 - r^1$, and one eigenvalue as $\lambda_1 = 1 + (C_0 - 1)r^1 = (1 - r^1) + r^1/C_0^{-1}$.

Let us now assume that the theorem is true for the balanced tree whose root node is at height $H - 1$. Then if we have a tree with height $H$. We call the resulting matrix $K_H$.

By the first bullet point of Theorem A.1 we directly get $\lambda_0$ from (i). Then by the second bullet point of Theorem A.1, the rest of the eigenvalues are from the symmetric matrix $A_{H-1} \in \mathbb{R}^{C_1 \times C_1}$ whose diagonal elements are $\gamma = 1 + (C_0/C_1 - 1)r^1$ and whose off diagonal elements are $C_0/C_1 \cdot r^j$ for $j \geq 2$.

The key is to observe that $A_{H-1}$ is still a block structured matrix. After $A_{H-1}$ is scaled by $\gamma$, the resulting matrix can be also seen as a result of maximizing CPCC where the off diagonal blocks have smaller values.

Applying the hypothesis induction, we then know the expression of eigenvalues for $A_{H-1}$ as

(i) we have $C_1 - C_2$ eigenvalues of the form

$$\lambda_1 = \gamma(1 - \frac{r^2 C_0/C_1}{\gamma})$$

$$= \gamma - \frac{C_0}{C_1} r^2$$

$$= 1 + (\frac{C_0}{C_1} - 1)r^1 - \frac{C_0}{C_1} r^2$$

$$= 1 - r^1 + \frac{C_0}{C_1}(r^1 - r^2)$$

$$= \lambda_0 + \frac{C_0}{C_1}(r^1 - r^2)$$

(ii) For $0 < h \le H - 2$, we have $C_{h+1} - C_{h+2}$ eigenvalues of the form

$$\lambda_{h+1} - \lambda_h = \gamma \frac{C_0}{C_1} \left( \frac{r^{h+1}}{\gamma} - \frac{r^{h+2}}{\gamma} \right) \frac{C_1}{C_{h+1}}$$

$$\lambda_{h+1} = \lambda_h + (r^{h+1} - r^{h+2}) \frac{C_0}{C_{h+1}}$$

(iii) The last eigenvalue is

$$\lambda_H - \lambda_{H-1} = C_1 r^H / \gamma \cdot \frac{C_0}{C_1} \gamma$$

$$= C_0 r^H$$

Therefore, we proved the theorem by showing the induction step from $K_{H-1}$ to $K_H$ holds. ∎

Note that the true symmetric covariance matrix $K'$ might not be having the exact format as $K$, but it can be seen as a perturbation of $K$ where $\|K' - K\| \le \epsilon$, $\epsilon$ is a small constant. By Weyl's inequality [93], the approximation error of each eigenvalue is bounded by $[\lambda_i - \epsilon, \lambda_i + \epsilon]$.

## B  Additional Algorithm and Experimental Details

In this section, we first provide an overview of our algorithm, followed by a discussion on the choice of the *flat* loss and additional experimental details about the training and evaluation metrics.

### B.1  Broader Impact Statement

Our work proposes `HypStructure`, a structured hyperbolic regularization approach to embed hierarchical information into the learnt representations. This provides significant advancements in understanding and utilizing hierarchical real-world data, particularly for tasks such as representation learning, classification and OOD detection, and we recognize both positive societal impacts and potential risks of this work. The ability to better model hierarchical data in a structured and interpretable fashion is particularly helpful for domains such as AI for science and healthcare, where the learnt representations will be more reflective of the underlying relationships in the domain space. Additionally, the low-dimensional capabilities of hyperbolic geometry can lead to gains in computational efficiency and reduce the carbon footprint in large scale machine learning. However, real-world hierarchical data often incorporates existing biases which may be amplified by structured representation learning, and hence it is important to incorporate fairness constraints to mitigate this risk.

### B.2  Pseudocode for `HypStructure`

The training scheme for our `HypStructure` based structured regularization framework is provided in Algorithm 1. At a high level, in `HypStructure`, we optimize a combination of the following two losses: (1) a *hyperbolic CPCC* loss to encourage the representations in the hyperbolic space to correspond with the label hierarchy, (2) a *hyperbolic centering* loss to position the representation corresponding to the root of the node at the centre of the Poincaré ball and the children nodes around it.

---

**Algorithm 1** HypStructure: Hyperbolic Structured Representation Learning

---

**Input:** Batch size $B$, Label tree $\mathcal{T} = (V, E, e)$, Number of epochs $K$, Task Loss formulation $\ell_{\text{Flat}}$,
Encoder $f_\theta$, Classifier Head $g_w$, Learning Rate $\eta$, Hyperparameters $\alpha, \beta$
1: Initialize model parameters: $\theta, w$
2: **for** epoch = $1, 2, \ldots K$ **do**
3:   **for** batch = $1, 2, \ldots, B$ **do**
4:    Get image-label pairs: $\{(\boldsymbol{x}_i, y_i)\}_{i=1}^B$
5:    Forward pass to compute the representations: $(\boldsymbol{z}_1 \ldots \boldsymbol{z}_B) \leftarrow (f_\theta(\boldsymbol{x}_1) \ldots (f_\theta(\boldsymbol{x}_B))$
6:    Compute the Task loss: $\ell_{\text{Flat}}(g_w(\boldsymbol{z}_i), y_i)$
7:    Get hyperbolic representations using exp. map (eq. (4)): $\tilde{\boldsymbol{z}}_i \leftarrow \exp_{\boldsymbol{0}}^c(\boldsymbol{z}_i)$
8:    Calculate class prototypes using hyp. Averaging (eq. (6)): $\omega_i \leftarrow \text{HypAve}_K(\tilde{\boldsymbol{z}}_1^v, \ldots \tilde{\boldsymbol{z}}_j^v)$
9:    Compute pairwise hyp. distances (eq. (3)) $\forall v_i, v_j \in V : \rho(v_i, v_j) \leftarrow d_{\mathbb{B}_c}(\omega_i, \omega_j)$
10:    Get hyp. CPCC loss (eq. (1)): $\text{HypCPCC}(d_\mathcal{T}, \rho)$
11:    Compute hyp. centering loss using (Equation (6)): $\ell_{\text{center}} = \|\text{HypAve}_B(\tilde{\boldsymbol{z}}_1, \ldots, \tilde{\boldsymbol{z}}_B)\|$
12:    Get total loss using Equation (8): $\mathcal{L}(\mathcal{D}_B)$
13:    Compute Gradients for learnable parameters at time $t$: $\mathbf{u_t}(\theta, w) \leftarrow \nabla_{\theta, w} \mathcal{L}(\mathcal{D}_B)$
14:    Refresh the parameters: $(\theta, w)_{t+1} \leftarrow (\theta, w)_t - \frac{\eta}{B} \mathbf{u_t}(\theta, w)$
**Output:** $(\boldsymbol{z}_1, \ldots \boldsymbol{z}_N); \theta, w$

---

## B.3 Choice of Flat loss

We use the Supervised Contrastive [39] (SupCon) loss as the first choice for a *flat* loss in our experimentation. Let $q_y$ be the one-hot vector with the $y$-th index as 1. The Cross Entropy (CE) loss, defined between the predictions $g \circ f_\theta(\boldsymbol{x})$ and the labels $y$, as $\ell_{\text{CE}}(g \circ f(\boldsymbol{x}), y) := -\sum_{i \in [k]} q_i \log(g(f(\boldsymbol{x}))_i)$ has been used quite extensively in large-scale classification problems in the literature [10, 11, 43, 98]. However, several shortcoming of the CE loss, such as lack of robustness [81, 107] and poor generalization [17, 54] have been discovered in recent research. Contrastive learning has emerged as a viable alternative to the CE loss, to address these shortcomings [7, 95, 29, 87, 33, 27]. The underlying principle for these methods is to *pull* together embeddings for positive pairs and *push* apart the embeddings for negative samples, in the feature space. In the absence of labels, positive samples are created by data augmentations of images and negative samples are randomly chosen from the minibatch. However, when the labels are available, the class information can be leveraged to extend this methodology as a Supervised Contrastive loss (SupCon) by *pulling* together embeddings from the same class, and *pushing* apart the embeddings from different classes. This offers a more stable solution for a variety of tasks [39, 76, 83].

**Definition B.1** (SupCon Loss). Given a training sample $\boldsymbol{x}_i$, feature encoder $f_\theta(\cdot)$ and a projection head $h(\cdot)$, we denote the normalized feature representations from the projection head as:

$$u_i = \frac{h(f_\theta(\boldsymbol{x}_i))}{\|h(f_\theta(\boldsymbol{x}_i))\|_2}, \tag{10}$$

For the $N$ training samples $\{(\boldsymbol{x}_i, y_i)\}_{i=1}^N$, we denote the training batch of $2N$ (augmented) pairs as $\{(\tilde{\boldsymbol{x}}_i, \tilde{y}_i)\}_{i=1}^{2N}$ and define the SupCon loss as:

$$\ell_{\text{SupCon}} = \frac{1}{2N} \sum_{i=1}^{2N} -\log \frac{\frac{1}{2N_{y_i}-1} \sum_{k=1}^{2N} \mathbb{1}(k \neq i) \mathbb{1}(\tilde{y}_k = \tilde{y}_i) e^{\boldsymbol{u}_i^T \cdot \boldsymbol{u}_k / \tau}}{\sum_{k=1}^{2N} \mathbb{1}(k \neq i) e^{\boldsymbol{u}_i^T \cdot \boldsymbol{u}_k / \tau}}, \tag{11}$$

where $N_{y_i}$ refers to the number of images with label $y_i$ in the batch, $\tau$ is the temperature parameter, $\cdot$ refers to the inner product, and $\boldsymbol{u}_i$ and $\boldsymbol{u}_k$ are the normalized feature representations using Equation (10) for $\tilde{\boldsymbol{x}}_i$ and $\tilde{\boldsymbol{x}}_k$ respectively.

While the numerator in the formulation in Equation (11) only considers the samples (and its augmentations) belonging to the same class, the denominator sums over all the negatives as well. Overall, this encourages the network to closely align the feature representations for *all* the samples belonging to the same class, while *pushing* apart the representations of samples across different classes.

We note that our proposed method HypStructure is not limited to the choice of euclidean classification losses as $\ell_{\text{Flat}}$ and we report additional results with hyperbolic classification losses in Sections C.8 and C.9 respectively, demonstrating the wide applicability of our approach.

## B.4    Implementation Details

### B.4.1    Software and Hardware

We implement our method in PyTorch 2.2.2 and run all experiments on a single NVIDIA GeForce RTX-A6000 GPU. The code for our methodology will be open sourced for a wider audience upon acceptance, in the spirit of reproducible research.

### B.4.2    Architecture, Hyperparameters and Training

We use the ResNet-18 [28] network as the backbone for CIFAR10, and ResNet-34 as the backbone for CIFAR100 and ImageNet100 datasets. We use a ReLU activated multi layer perceptron with one hidden layer as the projection head $h(.)$ where its hidden layer dimension is the same as input dimension size and the output dimension is 128. We follow the original hyperparameter settings from [39] for training the CIFAR10 and CIFAR100 models from scratch with a temperature $\tau = 0.1$, feature dimension 512, and training for 500 epochs with an initial learning rate of 0.5 with cosine annealing, optimizing using SGD with momentum 0.9 and weight decay $10^{-4}$, and a batch size of 512 for all the experiments. For ImageNet100, we finetune the ResNet-34 for 20 epochs following [61] with an initial learning rate of 0.01 and update the weights of the last residual block and the nonlinear projection head, while freezing the parameters in the first three residual blocks. We use the same $\alpha$ values as the regularization parameters for the CPCC loss in Equation (2) ($\ell_2$-CPCC) and in Equation (8) (our proposed method `HypStructure`) for a fair comparison and find that the default regularization hyperparameter for the CPCC loss $\alpha = 1.0$ for both $\ell_2$-CPCC and `HypStructure` performs well for the experiments on the CIFAR10 and CIFAR100 datasets. We observe that the experiments on the IMAGENET100 dataset benefit from a lower $\alpha = 0.5$. Additionally, we set the hyperparameter for the centering loss in our methodology as $\beta = 0.01$ for all the experiments. We use the default curvature value of $c = 1.0$ for the mapping and distance computations in the Poincaré ball.

### B.4.3    Datasets and Hierarchy

We use the following three datasets for our primary experimentation and training in this work

1. **CIFAR10** ([45]). It consists of 50,000 training images and 10,000 test images from 10 different classes.

2. **CIFAR100**([45]). It also consists of 50,000 training images and 10,000 test images, however the images belong to 100 classes. Note that the classes are not identical to the CIFAR10 dataset.

3. **ImageNet100**([72]). This dataset is created as a subset of the large-scale ImageNet dataset following [59]. The original ImageNet dataset consists of 1,000 classes and 1.2 million training images and 50,000 validation images. We construct the ImageNet100 dataset from this original dataset by sampling 100 classes, which results in 128,241 training images and 5000 validation images. We mention the specific classes used for sampling below.

Following [59], we use the below 100 class id's for creating the ImageNet100 subset: n03877845, n03000684, n03110669, n03710721, n02825657, n02113186, n01817953, n04239074, n02002556, n04356056, n03187595, n03355925, n03125729, n02058221, n01580077, n03016953, n02843684, n04371430, n01944390, n03887697, n04037443, n02493793, n01518878, n03840681, n04179913, n01871265, n03866082, n03180011, n01910747, n03388549, n03908714, n01855032, n02134084, n03400231, n04483307, n03721384, n02033041, n01775062, n02808304, n13052670, n01601694, n04136333, n03272562, n03895866, n03995372, n06785654, n02111889, n03447721, n03666591, n04376876, n03929855, n02128757, n02326432, n07614500, n01695060, n02484975, n02105412, n04090263, n03127925, n04550184, n04606251, n02488702, n03404251, n03633091, n02091635, n03457902, n02233338, n02483362, n04461696, n02871525, n01689811, n01498041, n02107312, n01632458, n03394916, n04147183, n04418357, n03218198, n01917289, n02102318, n02088364, n09835506, n02095570, n03982430, n04041544, n04562935, n03933933, n01843065, n02128925, n02480495, n03425413, n03935335, n02971356, n02124075, n07714571, n03133878, n02097130, n02113799, n09399592, n03594945.

In addition to the training and validation images, we also require the label hierarchy for each of these datasets for the CPCC computation in $\ell_2$-CPCC and `HypStructure` approaches. For CIFAR100, we use the three-level hierarchy provided with the dataset release[3]. We show this hierarchy in Table 3, where the top-level is the root of the tree.

Table 3: Class Hierarchy of the CIFAR100 Dataset

| Coarse Classes | Fine Classes |
| --- | --- |
| aquatic mammals | beaver, dolphin, otter, seal, whale |
| fish | aquarium fish, flatfish, ray, shark, trout |
| flowers | orchids, poppies, roses, sunflowers, tulips |
| food containers | bottles, bowls, cans, cups, plates |
| fruit and vegetables | apples, mushrooms, oranges, pears, sweet peppers |
| household electrical devices | clock, computer keyboard, lamp, telephone, television |
| household furniture | bed, chair, couch, table, wardrobe |
| insects | bee, beetle, butterfly, caterpillar, cockroach |
| large carnivores | bear, leopard, lion, tiger, wolf |
| large man-made outdoor things | bridge, castle, house, road, skyscraper |
| large natural outdoor scenes | cloud, forest, mountain, plain, sea |
| large omnivores and herbivores | camel, cattle, chimpanzee, elephant, kangaroo |
| medium-sized mammals | fox, porcupine, possum, raccoon, skunk |
| non-insect invertebrates | crab, lobster, snail, spider, worm |
| people | baby, boy, girl, man, woman |
| reptiles | crocodile, dinosaur, lizard, snake, turtle |
| small mammals | hamster, mouse, rabbit, shrew, squirrel |
| trees | maple, oak, palm, pine, willow |
| vehicles 1 | bicycle, bus, motorcycle, pickup truck, train |
| vehicles 2 | lawn-mower, rocket, streetcar, tank, tractor |

Since no hierarchy is available for the CIFAR10 and ImageNet100 datasets, we construct a hierarchy for CIFAR10 manually, as seen in Figure 2. For ImageNet100, we create a subtree from the WordNet[4] hierarchy, given the 100 aforementioned classes as leaves. We consider the classes which are one level above the leaf nodes in the hierarchy as the coarse classes, following [104].

For the task of OOD detection, we use the following five diverse OOD datasets for CIFAR10 and CIFAR100 as ID datasets, following the literature [83]: SVHN [65], `Textures` [9], `Places365` [109], `LSUN` [102] and `iSUN` [99]. When ImageNet100 is used as the ID dataset, we use 4 diverse OOD datasets as the ones in [37], namely subsets of `iNaturalist` [90], `SUN` [96], `Places` [109] and `Textures` [9]. These datasets have been processed so that there is no overlap with the ImageNet classes.

## B.5 Delta Hyperbolicity Metrics

We use Gromov's $\delta_{rel}$ to evaluate the tree-likeness of the data in Section 4.1, following [40]. For an arbitrary metric space $\mathcal{X}$ with metric $d$, for any three points $\boldsymbol{x}, \boldsymbol{y}, \boldsymbol{z} \in \mathcal{X}$, we can define the Gromov product as

$$(\boldsymbol{y}, \boldsymbol{z})_{\boldsymbol{x}} = \frac{1}{2}(d(\boldsymbol{x}, \boldsymbol{y}) + d(\boldsymbol{x}, \boldsymbol{z}) - d(\boldsymbol{y}, \boldsymbol{z}))$$

Then, $\delta$-hyperbolicity can be defined as the minimum value of $\delta$ such that for any four points $\boldsymbol{x}, \boldsymbol{y}, \boldsymbol{z}, \boldsymbol{w} \in \mathcal{X}$, the following condition holds:

$$(\boldsymbol{x}, \boldsymbol{z})_{\boldsymbol{w}} \geq \min((\boldsymbol{x}, \boldsymbol{y})_{\boldsymbol{w}}, (\boldsymbol{y}, \boldsymbol{z})_{\boldsymbol{w}}) - \delta$$

It can be shown that equivalently, there exists a geometric definition of $\delta$-hyperbolicity. A geodesic triangle in $\mathcal{X}$ is $\delta$-slim if each of its side is contained in the $\delta$-neighbourhood of the union of the other two sides. We define $\delta$-hyperbolicity as the minimum $\delta$ that guarantees any triangle in $\mathcal{X}$ is $\delta$-slim. From Figure 10, when the curvature of the surface increases, the geodesic triangle converges to a tree/star graph, and $\delta$ gradually reduces to 0.

Following [40], we use the scale-invariant metric $\delta_{rel} = \frac{2\delta}{\text{diam}(\mathcal{X})}$ for evaluation, so that the $\delta_{rel}$ is normalized in $[0, 1]$, and the $\text{diam}(\cdot)$ is the set diameter or the maximal pairwise distance.

---

[3]https://www.cs.toronto.edu/ kriz/cifar.html
[4]https://www.nltk.org/howto/wordnet.html

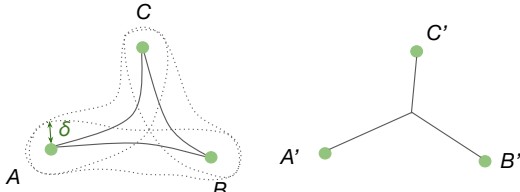

Figure 10: Example of a $\delta$-slim triangle, where each side of $\triangle ABC$ is the geodesic distance of two points in the metric space.

Table 4: Linear classification accuracy using `SupCon` [39] as $\ell_{\text{Flat}}$.

| Dataset | Method (SupCon) | Fine Accuracy ($\uparrow$) |
|---|---|---|
| CIFAR10 | Flat | 94.53 |
| | $\ell_2$-CPCC | 95.08 |
| | HypStructure (Ours) | **95.18** |
| CIFAR100 | Flat | 75.11 |
| | $\ell_2$-CPCC | 75.66 |
| | HypStructure (Ours) | **77.66** |

## C  Additional Experimental Results

### C.1  Results using Linear Evaluation

We also perform an evaluation of the *fine* classification accuracy following the common linear evaluation protocol [39] where a linear classifier is trained on top of the normalized penultimate layer features. We report these accuracies for the models trained on the CIFAR10 and CIFAR100 datasets in Table 4 for the leaf-only variants of the models. We observe that the relative trend of accuracies is identical to the ones reported using the kNN evaluation in Table 1 and our proposed method `HypStructure` outperforms the *flat* and $\ell_2$-CPCC methods on both the datasets.

### C.2  Component-wise Ablation Study of `HypStructure`

To understand the role of each component in our proposed methodology `HypStructure`, we perform a detailed ablation study with the different components and measure the *fine* and the *coarse* accuracies on the CIFAR100 dataset. Specifically, we examine

1. the role of embedding all internal nodes in the label hierarchy (eq. (8) and line 10 in Algorithm 1), as opposed to only using leaf nodes as in [104]. We refer to the inclusion of internal nodes as $\mathcal{T}_{\text{int}}$.

2. the role of hyperbolic class centroids computation using hyperbolic averaging (eq. (6) and line 8 in Algorithm 1), as opposed to the Euclidean computation of class prototypes as in [104]. We refer to the hyperbolic class centroid computation as $\omega_{\text{hyp}}$.

3. the role of the hyperbolic centering loss in our proposed methodology (eq. (8) and line 11 in Algorithm 1), as opposed to not using a centering loss. We refer to the inclusion of the centering loss as $\ell_{\text{center}}$.

We ablate over the aforementioned settings, where a $\checkmark$ denotes the inclusion of that setting, and report the results on the CIFAR100 dataset in Table 5. Firstly, we observe that while the centering loss $\ell_{\text{center}}$ improves the coarse accuracy only by a small increment, it leads to a significant improvement in the fine accuracy (rows $1 \rightarrow 2$ and $4 \rightarrow 5$), indicating that the centering of the root in the poincare disk allows for a better relative positioning of the fine classes within the coarse class groups. Secondly, we observe that both the inclusion of internal nodes $\mathcal{T}_{\text{int}}$, and the hyperbolic computation of the class centroids $\omega_{\text{hyp}}$ is critical for accurately embedding the hierarchy, and removing either of these components (i.e. rows $5 \rightarrow 3$ for $\mathcal{T}_{\text{int}}$ and rows $5 \rightarrow 2$ for $\omega_{\text{hyp}}$), leads to a degradation in both the fine as well as the coarse accuracies. The best overall performance is observed when all three of the components are included (row 5).

Table 5: Ablation study on the components of `HypStructure`. We report the Classification accuracies based on the CIFAR100 model trained with ResNet-34.

| HypStructure **Components** | | | **Classification Acc.↑** | |
| --- | --- | --- | --- | --- |
| **Internal Nodes ($\mathcal{T}_{\text{int}}$)** | **Hyp. Class Centroids ($\omega_{\text{hyp}}$)** | **Hyp. Centering ($\ell_{\text{center}}$)** | **Fine** | **Coarse** |
| ✓ | | | 75.03 | 84.77 |
| ✓ | | ✓ | 75.61 | 84.81 |
| | ✓ | ✓ | 76.22 | 85.70 |
| ✓ | ✓ | | 76.59 | **86.23** |
| ✓ | ✓ | ✓ | **76.91** | 86.22 |

## C.3    OOD detection

### C.3.1    Related Work and Methods

The goal of prior works in the OOD literature is the supervised setting of learning an accurate classifier for ID data, along with an ID-OOD detection methodology and this task has been explored in the generative model setting [42, 63, 71, 77, 97], and more extensively in the supervised discriminative model setting [2, 30, 36, 37, 51, 53, 82, 60]. The methods in this setting can be categorized into four sub-categories following [106], primarily:

**Post-Hoc Inference**    These methods design post-processing/scoring mechanisms on base classifiers such as MSP [30], ODIN [51], ReAct [82], SSD+ [76], KNN+ [83] and RankFeat [80].

**Training without outlier data**    These methods involve training-time regularization or different objective functions for improving OOD detection capabilities such as G-ODIN [36], CSI [85], LogitNorm [92] and CIDER [61].

**Training with outlier data**    These methods assume access to auxiliary OOD training samples such as OE [31] and MixOE [105].

**Data Augmentation**    These methods improve the generalization ability of image classifiers such as StyleAugment [22], AugMix [32] and RegMixup [69].

Our proposed work can be considered primarily in the **Training without outlier data** category, and we note that none of the prior works use any additional structural regularization term in the objective functions.

### C.3.2    Dataset-wise OOD Detection Results

Table 6: Results on CIFAR10. OOD detection performance for ResNet-18 trained on CIFAR10. Training with `HypStructure` achieves strong OOD detection performance.

| Method | OOD Dataset AUROC (↑) | | | | | Avg. (↑) |
| --- | --- | --- | --- | --- | --- | --- |
| | SVHN | Textures | Places365 | LSUN | iSUN | |
| ProxyAnchor | 94.55 | 93.16 | 92.06 | 97.02 | 96.56 | 94.67 |
| CE + SimCLR | 99.22 | 96.56 | 86.70 | 85.60 | 86.78 | 90.97 |
| CSI | 94.69 | 94.87 | 93.04 | 97.93 | 98.01 | 95.71 |
| CIDER | 99.72 | 96.85 | 94.09 | 99.01 | **96.64** | 97.26 |
| SSD+ | 99.51 | 98.35 | **95.57** | 97.83 | 95.67 | 97.38 |
| KNN+ | 99.61 | 97.43 | 94.88 | 98.01 | 96.21 | 97.22 |
| $\ell_2$-CPCC | 93.27 | 94.76 | 60.15 | 75.29 | 59.87 | 76.67 |
| HypStructure (Ours) | **99.75** | **98.89** | 94.80 | **99.67** | 95.64 | **97.75** |

We report the dataset-wise OOD detection results in Tables 7a, 6 and 7 for CIFAR100, CIFAR10 and ImageNet100 respectively. We compare with several other state-of-the-art baseline OOD detection

Table 7: Results on ImageNet100. OOD detection performance for ResNet-34 trained on ImageNet100. Training with `HypStructure` achieves strong OOD detection performance.

| Method | OOD Dataset AUROC (↑) | | | | Avg. (↑) |
|---|---|---|---|---|---|
| | SUN | Places365 | Textures | iNaturalist | |
| CIDER | 91.63 | 89.29 | 97.98 | 96.35 | 93.81 |
| SSD+ | 88.97 | 85.98 | **98.49** | 96.42 | 92.46 |
| KNN+ | 89.48 | 86.64 | 98.38 | **96.46** | 92.74 |
| $\ell_2$-CPCC | 90.95 | 86.87 | 97.41 | 90.08 | 91.33 |
| `HypStructure` (Ours) | **92.21** | **90.12** | 97.33 | 95.61 | **93.83** |

methods for CIFAR10 and CIFAR100, namely ProxyAnchor [41], SimCLR [7] CSI [85], and CIDER [61] respectively. Results for these methods are taken from CIDER [61] where contrastive learning based OOD detection methods typically outperforms non-contrastive learning ones. For ImageNet100, in the absence of the available class ids used to train the original models in CIDER [61], we finetune the ResNet34 models on the created ImageNet100 dataset. For CIDER and SupCon, we use the official implementations and hyperparameters provided by the authors.

We observe that our proposed method leads to an improvement in the average OOD detection AUROC over all the ID datasets. In practice, we find that the Euclidean-centroid computational variant (first compute the Euclidean centroids and then apply the exponential map) of our proposed method performs slightly better than the hyperbolic-centroid computational variant (first apply the exponential map and then compute the hyperbolic average), for the specific task of OOD detection, while having equivalent performance on the ID classification task. Hence, we report the OOD detection accuracy corresponding to the first version.

## C.4 Visualization of Learned Features

We provide additional visualizations of the learnt features from our proposed method `HypStructure` on the CIFAR10, CIFAR100 and ImageNet100 datasets in Figures 11, 12 and 13 respectively.

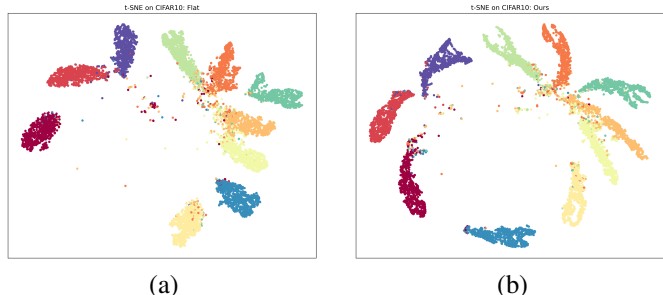

(a)                                   (b)

Figure 11: Euclidean t-SNE Visualizations on CIFAR10.

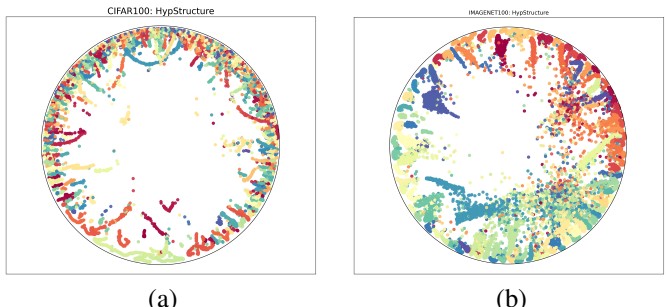

(a)                                   (b)

Figure 12: Hyperbolic UMAP Visualizations on CIFAR100 and ImageNet100.

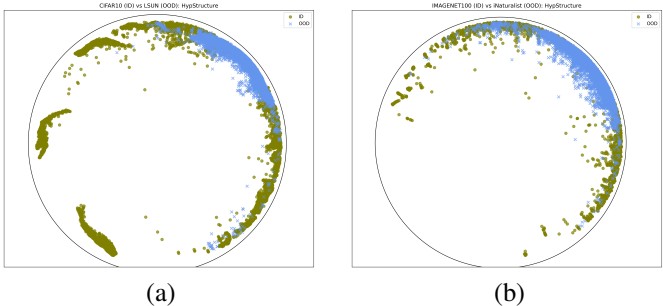

(a)                  (b)

Figure 13: Hyperbolic UMAP Visualizations of ID-OOD separation on CIFAR10 and ImageNet100.

## C.5   Effect of Centering Loss and Embedding Internal Node

Embedding the internal tree nodes in `HypStructure` $\mathcal{T}_{\text{int}}$ (as compared to only leaf nodes in prior work CPCC) and placing the root node at the center of the Poincaré disk with $\ell_{\text{center}}$ loss, helps in embedding the hierarchy more accurately. To understand the impact of these components, we first visualize the learnt representations from `HypStructure`, with and without these components - i.e. embedding internal nodes and a centering loss vs leaf only nodes, via UMAP in Figure 14 (CIFAR100) and Figure 15 (ImageNet100). We also provide a performance comparison (fine accuracy) in Table 8.

| Method | CIFAR10 | CIFAR100 | ImageNet100 |
|---|---|---|---|
| `HypStructure` (leaf only) | 94.54 | 76.22 | 89.85 |
| `HypStructure` (with internal nodes and centering) | **94.79** | **76.68** | **90.12** |

Table 8: Fine accuracy comparison of `HypStructure` with vs. without internal nodes and centering on CIFAR10, CIFAR100, and ImageNet100 datasets.

First, based on Figures Figure 14 and Figure 15, one can note that in the leaf-only setting without embedding internal nodes and centering loss (figures on the left), the samples belonging to the fine classes which share the same parent (same color) are in close proximity reflecting the hierarchy accurately, however the samples are not spread evenly. With the embedding of internal nodes and a centering loss (right), we note that the representations are spread between the center (root) to the boundary as well as across the Poincaré disk, which is more representative of the original hierarchy. This also leads to performance improvements as can be seen in Table 8.

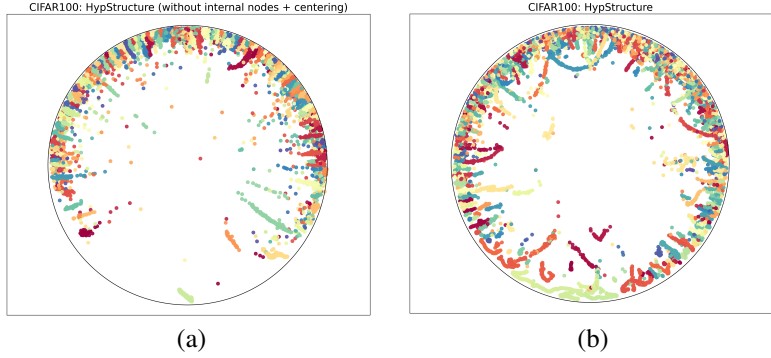

(a)                  (b)

Figure 14: Hyperbolic UMAP Visualizations on CIFAR100 using `HypStructure` without embedding the internal nodes and a hyperbolic centering loss (left), and with embedding the internal nodes along with a centering loss (right).

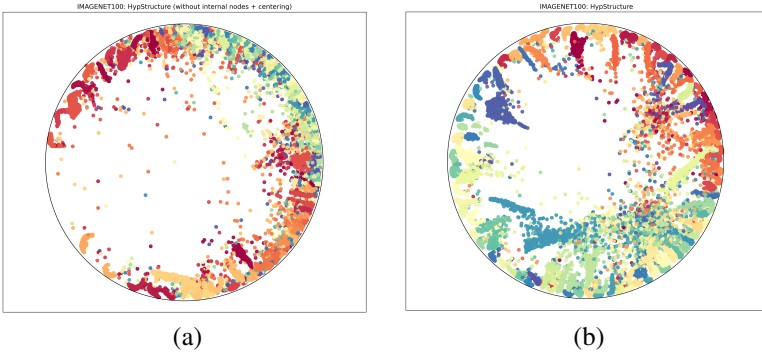

(a)                                                        (b)

Figure 15: Hyperbolic UMAP Visualizations on ImageNet100 using `HypStructure` without embedding the internal nodes and a hyperbolic centering loss (left), and with embedding the internal nodes along with a centering loss (right).

## C.6   Effect of Label Hierarchy Weights

Compared to ranking-based hyperbolic losses [66], our HypCPCC factors in absolute values of the node-to-node distances. The learned hierarchy with HypCPCC will not only implicitly encode the correct parent-child relations, but can also learn more complex and weighted hierarchical relationships more accurately. To demonstrate this, we modify the CIFAR10 tree hierarchy, and gradually increase the weight for the left transportation branch to $2\times$ and $4\times$ and use new weighted trees for the CPCC tree distance computation. We visualize the learnt representations in Figure 16 and we can observe that in the learned representations from left to right, the distance between the transportation classes (blue) are larger as compared to other classes, as expected.

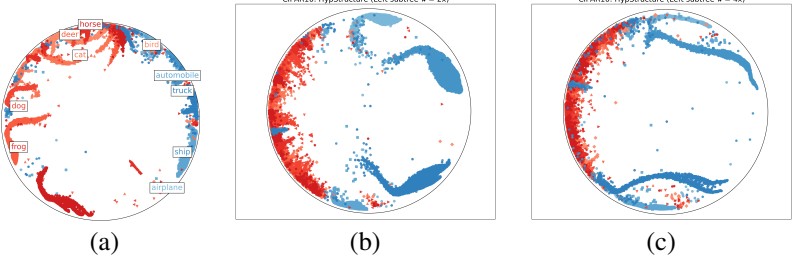

(a)                                  (b)                                  (c)

Figure 16: HypStructure can learn more nuanced representations with weighted hierarchy trees. Hyperbolic UMAP visualizations on CIFAR10 using HypStructure with differently weighted left-subtrees.

## C.7   Effect of Klein Averaging

We experiment with the two HypCPCC variants, using Klein Averaging or Euclidean mean for centroid computation, as mentioned in Section 3.2 and report the results in Table 9. We empirically observe that using the Klein averaging leads to performance improvements across datasets.

| Method | CIFAR10 | CIFAR100 | ImageNet100 |
|---|---|---|---|
| Euclidean | 94.56 | 75.64 | 90.08 |
| Klein | **94.79** | **76.68** | **90.12** |

Table 9: Fine accuracy comparison between Euclidean and Klein centroid computation methods in HypCPCC on CIFAR10, CIFAR100, and ImageNet100 datasets.

## C.8 Experiments with the Hyperbolic Supervised Contrastive Loss

We experiment with the Hyperbolic Supervised Contrastive Loss as proposed in [21] as the choice of the $\ell_{\text{Flat}}$ loss and refer to this loss as $\ell_{\text{HypSupCon}}$. We follow the original setup as described by the authors for the measurement of the $\ell_{\text{HypSupCon}}$, where the representations from the encoders are not normalized directly, instead an exponential map is used to project these features from the Euclidean space to the Poincaré ball first. Then, the inner product measurement in the $\ell_{\text{SupCon}}$ is replaced with the negative hyperbolic distances in the Poincaré ball to compute the $\ell_{\text{HypSupCon}}$ loss. We also experiment with our proposed methodology HypStructure along with the $\ell_{\text{HypSupCon}}$ loss and report the classification accuracies and hierarchy embedding metrics for both these settings in Table 10. We further report the OOD detection performance on CIFAR10, CIFAR100 and ImageNet100 as in-distribution datasets for both of these settings in Tables 11, 12 and 13 respectively. We observe that using HypStructure with a hyperbolic loss such as $\ell_{\text{HypSupCon}}$ as the Flat loss leads to improvements in accuracy across classification and OOD detection tasks while also improving the quality of embedding the hierarchy. This demonstrates the wide applicability of our proposed method HypStructure which can be used in conjunction with both euclidean and non-euclidean classification losses.

Table 10: Evaluation of hierarchical information distortion and classification accuracy using HypSupCon [21] as $\ell_{\text{Flat}}$. All metrics are reported as mean (standard deviation) over 3 seeds.

| Dataset (Backbone) | Method | Distortion of Hierarchy | | Classification Accuracy | |
|---|---|---|---|---|---|
| | | $\delta_{rel}$ ($\downarrow$) | CPCC ($\uparrow$) | Fine ($\uparrow$) | Coarse ($\uparrow$) |
| CIFAR10 (ResNet-18) | Flat | 0.128 (0.007) | 0.745 (0.017) | 94.58 (0.04) | 98.96 (0.01) |
| | HypStructure | **0.017 (0.001)** | **0.989 (0.001)** | **95.04 (0.02)** | **99.36 (0.02)** |
| CIFAR100 (ResNet-34) | Flat | 0.168 (0.002) | 0.664 (0.012) | 75.81 (0.06) | 85.26 (0.07) |
| | HypStructure | **0.112 (0.005)** | **0.773 (0.008)** | **76.22 (0.14)** | **85.83 (0.06)** |
| ImageNet100 (ResNet-34) | Flat | 0.157 (0.004) | 0.473 (0.004) | 89.87 (0.01) | 90.41 (0.01) |
| | HypStructure | **0.126 (0.002)** | **0.714 (0.003)** | **90.26 (0.01)** | **90.95 (0.01)** |

Table 11: Results on CIFAR10 when using the HypSupCon[21] as $\ell_{\text{Flat}}$ using ResNet-18 as the backbone. Training with HypStructure achieves improvements in OOD detection performance.

| Method | OOD Dataset AUROC ($\uparrow$) | | | | | Avg. ($\uparrow$) |
|---|---|---|---|---|---|---|
| | SVHN | Textures | Places365 | LSUN | iSUN | |
| $\ell_{\text{HypSupCon}}$ | 89.45 | 93.39 | 90.18 | 98.18 | 91.31 | 92.51 |
| $\ell_{\text{HypSupCon}}$ + HypStructure (Ours) | **91.11** | **94.45** | **93.52** | **99.05** | **95.24** | **94.68** |

Table 12: Results on CIFAR100 when using the HypSupCon[21] as $\ell_{\text{Flat}}$ using ResNet-34 as the backbone. Training with HypStructure achieves improvements in OOD detection performance.

| Method | OOD Dataset AUROC ($\uparrow$) | | | | | Avg. ($\uparrow$) |
|---|---|---|---|---|---|---|
| | SVHN | Textures | Places365 | LSUN | iSUN | |
| $\ell_{\text{HypSupCon}}$ | 80.16 | 79.61 | 74.02 | 70.22 | 82.35 | 77.27 |
| $\ell_{\text{HypSupCon}}$ + HypStructure (Ours) | **82.28** | **83.51** | **77.95** | **86.64** | 69.86 | **80.05** |

## C.9 Experiments with a Hyperbolic Backbone

We experiment with Clipped Hyperbolic Neural Networks (HNNs) [26] as a hyperbolic backbone and use our proposed methodology HypStructure in conjunction with the hyperbolic Multinomial Logistic Regression (MLR) loss. We report the classification accuracies and hierarchy embedding metrics on the CIFAR10 and CIFAR100 datasets in Table 14, and the OOD detection performances using CIFAR10 and CIFAR100 as in-distribution datasets in Tables 15 and 16 respectively. We observe that using HypStructure along with a hyperbolic backbone leads to improvements in classification accuracies, reduced distortion in embedding the hierarchy, and improved OOD detection performance overall, demonstrating the wide applicability of HypStructure with hyperbolic networks.

Table 13: Results on ImageNet100 when using the HypSupCon[21] as $\ell_{\text{Flat}}$ using ResNet-34 as the backbone. Training with HypStructure achieves improvements in OOD detection performance.

| Method | OOD Dataset AUROC (↑) | | | | Avg. (↑) |
|---|---|---|---|---|---|
| | SUN | Places365 | Textures | iNaturalist | |
| $\ell_{\text{HypSupCon}}$ | 91.96 | 90.74 | **97.42** | 94.04 | 93.54 |
| $\ell_{\text{HypSupCon}}$ + HypStructure (Ours) | **93.87** | **91.56** | 97.04 | **95.16** | **94.41** |

Table 14: Evaluation of hierarchical information distortion and classification accuracy using Clipped Hyperbolic Neural Networks [26] as the backbone. All metrics are reported as mean (standard deviation) over 3 seeds.

| Dataset (Backbone) | Method | Distortion of Hierarchy | | Classification Accuracy | |
|---|---|---|---|---|---|
| | | $\delta_{rel}$ (↓) | CPCC (↑) | Fine (↑) | Coarse (↑) |
| CIFAR10 (Clipped HNN [26]) | Flat | 0.084 (0.008) | 0.604 (0.004) | 94.81 (0.23) | 89.71 (2.04) |
| | HypStructure | **0.013 (0.002)** | **0.988 (0.001)** | **94.97 (0.12)** | **98.35 (0.22)** |
| CIFAR100 (Clipped HNN [26]) | Flat | 0.098 (0.001) | 0.528 (0.009) | 76.46 (0.26) | 49.26 (0.73) |
| | HypStructure | **0.064 (0.006)** | **0.624 (0.005)** | **77.96 (0.14)** | **55.46 (0.61)** |

Table 15: Results on CIFAR10 when using the Clipped Hyperbolic Neural Networks [26] as the backbone. Training with HypStructure achieves improvements in OOD detection performance.

| Method | OOD Dataset AUROC (↑) | | | | | Avg. (↑) |
|---|---|---|---|---|---|---|
| | SVHN | Textures | Places365 | LSUN | iSUN | |
| Clipped HNN [26] | 92.63 | 90.74 | 88.46 | 95.66 | 92.41 | 91.98 |
| Clipped HNN [26] + HypStructure (Ours) | **95.41** | **93.91** | **92.31** | **96.87** | **94.92** | **94.68** |

Table 16: Results on CIFAR100 when using the Clipped Hyperbolic Neural Networks [26] as the backbone. Training with HypStructure achieves improvements in OOD detection performance.

| Method | OOD Dataset AUROC (↑) | | | | | Avg. (↑) |
|---|---|---|---|---|---|---|
| | SVHN | Textures | Places365 | LSUN | iSUN | |
| Clipped HNN [26] | 89.94 | 83.77 | 77.26 | 82.87 | 82.35 | 83.23 |
| Clipped HNN [26] + HypStructure (Ours) | **91.56** | **84.31** | **78.45** | **87.53** | **83.44** | **85.06** |

