# OpenReview forum: "Learning Structured Representations with Hyperbolic Embeddings"
_NeurIPS.cc/2024/Conference — NeurIPS 2024 poster_

### Official Review · Reviewer_3c64 · 2024-07-10

**Soundness:** 3
**Presentation:** 3
**Contribution:** 2
**Rating:** 5
**Confidence:** 3

**Summary:**

The paper introduces a novel regularization method, HypStructure, which utilizes hyperbolic geometry to improve the embedding of hierarchical relationships within feature representations. This approach enhances the learning of structured representations, reducing distortion and boosting generalization in low-dimensional scenarios. It also demonstrates superior performance in Out-of-Distribution (OOD) detection across various datasets through extensive empirical evaluation. Additionally, the paper includes an eigenvalue analysis that provides deeper insights into the structured representations, correlating positively with improved OOD detection performance. This advancement extends structured representation learning to hyperbolic spaces, achieving more discriminative and interpretable features that effectively capture the inherent hierarchies in complex datasets.

**Strengths:**

1. The paper is the first to formally characterize properties of hierarchy-informed features via an eigenvalue analysis, and also relate it to the OOD detection task.
2. The paper is easy to read and follow, making complex concepts accessible. The use of clear definitions, structured methodology sections, and detailed discussions helps in understanding both the theoretical underpinnings and practical implications of HypStructure. Visual aids and empirical results are presented in a manner that clearly supports the claims made.
3. The significance of this work lies in its potential impact on a range of applications that require understanding and leveraging hierarchical relationships in data, such as image recognition and OOD detection.

**Weaknesses:**

1. The main concern of this paper is the novelty. I believe the method proposed by the author in this work has been explored in many previous works. For instance, in "Hyperbolic Image Embeddings," "Hyperbolic Contrastive Learning for Visual Representations beyond Objects", etc. Although the paper characterizes properties of hierarchy-informed features via an eigenvalue analysis, the contribution is not significant enough to be accepted.
2. The writing is also not good enough for me. For instance, two examples starting from line 107 are not necessarily included in the formal paper; it is better to be placed in the supplementarity materials. There are also some repetitive expressions, for instance, in line 351 and line 353 (While the current work).
3. In summary, I believe the technical contribution of this paper is not significant enough to be accepted.

**Questions:**

See weaknesses.

**Limitations:**

As I said in the previous section, the main concern is the novelty. Potential improvements include extending the application scenarios of hyperbolic structured representations on more vision or language tasks that have not been explored before. Further theoretical investigation into establishing the error bounds of CPCC-style structured regularization objectives is of interest as well. The writing needs to be improved as well.

---

> ### Author Rebuttal · Authors · 2024-08-06
>
> We thank the reviewer for their detailed comments, and address each of the questions and weaknesses below.
>
> ## Concerns about the novelty
> We humbly disagree with the reviewers’ statement since our learning setting is **different from the existing works** in the hyperbolic geometry literature, where our focus is on using hyperbolic geometry to **embed a given class hierarchy.** We believe our work would be a suitable contribution to the space of structured representation learning and Hyperbolic geometry, and we will make key points to argue the novelty of our work, which will hopefully highlight our technical contributions to the field.
>
>
> 1. **Implicit vs explicit incorporation of hierarchy in Hyperbolic geometry and complementary nature of HypStructure**:  Most previous works in Hyperbolic geometry assume a latent **implicit** hierarchy in distributions/parameters, while our work focuses on an **explicit** knowledge injection of external hierarchy into the representation space. So while the methods and motivations may _seem_ similar, there are important differences in the nature of the problem, and the statement that the “method proposed by the author has been explored in many previous works” is an overly simplified generalization.
>
> Specifically, the “Hyperbolic Image Embeddings” paper replaces Euclidean layers with Hyperbolic classifiers which constrained the layer parameters in the Hyperbolic space, and has no similarities with HypStructure which instead utilizes external hierarchy for embedding regularization. Furthermore, the “Hyperbolic Contrastive Learning for Visual Representations Beyond Objects” proposes a Hyperbolic supervised contrastive loss implicitly assuming the existence of an unknown scene-object hierarchy.
>
> However, we want to emphasize that our approach offers a flexible framework that can be complementary and used in conjunction with other Hyperbolic losses/backbones assuming implicit hierarchies for performance gains. For instance, with a more recent variant of Hyperbolic classifier-based backbone (Clipped HNN), as well the Hyperbolic SupCon loss for further performance gains - please refer to the detailed results in the global response above, however we share these performance numbers here for reference.
>
> | Method | CIFAR10 | CIFAR100 | ImageNet100|
> |---|:---:|:---:|:---:|
> | L_hyp (HypSupCon)  | 94.64 | 75.77| 90.02|
> | L_hyp (HypSupCon) + HypStructure (Ours)| **95.06**| **76.08** | **90.31** |
>
> | Method | CIFAR10 | CIFAR100 |
> |---|:---:|:---:|
> | Clipped HNN | 95.08 | 77.42 |
> | Clipped HNN + HypStructure | **95.19** | **78.05** |
>
>
>
> 2. **Address limitations of the l2-CPCC with strong empirical performance**: The central theme of our paper is **accurately** embedding **external knowledge** available in the form of hierarchical information, in the representation space. While our work draws inspiration from recent contributions in Hyperbolic machine learning, we propose a novel, practical and effective framework that addresses the severe limitations of the CPCC objective in embedding this hierarchical information. This is clearly demonstrated with the strong empirical performance on large-scale datasets and visualization of learned representations using HypStructure.
>
> 3. **Theoretical understanding of structured representation learning and OOD detection** - to the best of our knowledge, our work is one of the first to draw theoretical connections between the paradigm of hierarchy-informed representation learning and tasks such as OOD detection, and we provide a provable hierarchical embedding that fills a theoretical gap in the broader field. These insights can serve to be useful in designing theoretically grounded, efficient and accurate representation learning methods that can learn general representations useful across tasks in practice as we have demonstrated for classification, OOD detection, and maintaining interpretable representations.
>
> Based on these grounds, we feel our work makes many novel contributions in terms of explicit modeling of structured hierarchy, improving and preserving representations with other Hyperbolic objectives and backbones, empirical effectiveness across tasks, and drawing theoretical connections to the OOD detection task (as you have rightly noted). We also foresee our contribution as motivating more works in the practical domain of embedding structured information explicitly using Hyperbolic geometry. We sincerely hope this provides a satisfactory justification to the reviewer regarding our contribution.
>
> ---
>
> ## Concerns about writing and examples
>
>
> We have included these examples in the motivation section of our paper to emphasize the practical concerns and severe limitations of l2-CPCC, which serves as our motivation to develop better methods. These examples related to our label tree setting, are very informative for understanding the problem setup and the challenges  (as has also been noted by Reviewer dBDC as a strength of our paper). Regarding writing, thanks for your suggestions, we will proofread to improve readability and remove repetitive expressions in the camera ready version.

---

> > ### Comment · Reviewer_3c64 · 2024-08-12
> >
> > Thanks for the authors' rebuttal. The rebuttal strengthens my understanding of this paper. Combined with other reviews and rebuttals. I still believe the plug-in approach is useful, but the improvements on CIFAR 10 and 100 are not obvious. However, the theoretical understanding part and mathematical proof are valuable to the community. I would like to raise the score to 5. Good luck!

---

> > > ### Author Response · Authors · 2024-08-12
> > > **Response to Reviewer 3c64**
> > >
> > > Thanks for raising your score and for the support of our work and it's contributions! We greatly appreciate the time you spent reviewing our paper, and going through the rebuttal and the other reviews, as well as your valuable suggestions for improving our work. We hope that the improvements across datasets, tasks and settings in terms of accuracies, distortion metrics and OOD scores are clearer in our updated set of results (as shared in response to reviewer gWWe, as well as the global comment with new results), and are grateful that our rebuttal could address your concerns. We would be happy to discuss further if you have any more questions about our work.

---

### Official Review · Reviewer_gWWe · 2024-07-11

**Soundness:** 2
**Presentation:** 3
**Contribution:** 3
**Rating:** 5
**Confidence:** 4

**Summary:**

The paper presents a novel approach, HypStructure, for learning structured representations. Comparing with the existing method, the proposed method adds an regularizer calculated from hyperbolic geometry. This approach aims to reduce distortion and improve generalization performance, particularly in low-dimensional scenarios. Extensive experiments are conducted on both the classification task and the out-of-distribution detection task.

**Strengths:**

The paper is well organized. The paper extends and studies the existing L2-CPCC to the hyperbolic space, which effectively reduces distortion and enhances the representation of hierarchical relationships in the data. The paper also conducted comprehensive experiments as well as detailed theoretical analysis of the eigenspectrum of structured representations.

**Weaknesses:**

If my understanding of the proposed loss term is correct, $L_flat$ is not calculated in the hyperbolic geometry. Have you tried the $L_flat$ with hyperbolic network or hyperbolic geometry. I hope the authors could provide more explanation of the combined loss in different geometries.

In Section 2.1, it mentions that $D_i$ is the subset of data points with a specific class label, and $d(D_i, D_j)$ is the distance between the feature vectors of $D_i$ and $D_j$. However, it is not mentioned how the vectors for $D_i$ and $D_j$ are calculated. Is it simply the average of all the feature vectors in the subset?

For Example 1 in Section 2.2, tree $T$ and nodes $G, C, D, E$ are referenced in a way that implies there should be a figure accompanied by the example. While Figure 1 is referenced shortly before this, it is meant to be accompanied by Example 2. Is there a figure that has been omitted here?

Also, I would recommend proofreading the paper to correct all grammatical errors. For example, in the paragraph of Section 2.1, the first sentence “Using a composite objective as defined in Equation (2), we can enforce the distance relationship between a pair of representations in the feature space, to behave similarly as the tree metric between the same vertices” should be corrected to “Using a composite objective as defined in Equation (2), we can enforce the distance relationship between a pair of representations in the feature space to behave similarly to the tree metric between the same vertices.” This version removes the unnecessary comma and corrects “behave similarly as…” to “behave similarly to…”.

**Questions:**

see my comments in the Weaknesses section.

**Limitations:**

There are two limitations mentioned in this paper. The first being that the L2-CPCC can only approximate the structured information since it is impossible to “embed a tree $T$ into $L2$ without distortion,” and as this work extends the L2-CPCC, it would similarly have this limitation. However, this is never explicitly stated in Section 7, where it mentions that HypStructure is only limited to Euclidean classification losses. There could have been more said about the limitations, but it does suffice.

---

> ### Author Rebuttal · Authors · 2024-08-06
>
> We thank the reviewer for their detailed comments, and address each of the questions and weaknesses below.
>
> ## Hyperbolic $L_{flat}$
>
> We choose the Supervised Contrastive Loss (SupCon) loss as the $L_{flat}$ loss in our experiments primarily to provide a fair comparison with the prior works, whereas our HypStructure methodology is flexible and can be combined with any other loss function.
>
> To demonstrate the wider applicability of our method based on the reviewers suggestion, we evaluate the performance using the Hyperbolic Supervised Contrastive loss $L_{\text{hyp}}$ proposed in [1] instead of the SupCon loss, and provide a comparison with using HypStructure in conjunction with this Hyperbolic loss in the Table below (fine acc.)
>
>
> | Method | CIFAR10 | CIFAR100 | ImageNet100|
> |---|:---:|:---:|:---:|
> | L_hyp (HypSupCon)  | 94.64 | 75.77| 90.02|
> | L_hyp (HypSupCon) + HypStructure (Ours)| **95.06**| **76.08** | **90.31** |
>
> In addition to the aforementioned loss, we vary the backbone network - the Clipped HNN model, a type of Hyperbolic Neural Network, instead of using the Euclidean ResNet style architectures and provide the results for that setting as below.
>
> | Method | CIFAR10 | CIFAR100 |
> |---|:---:|:---:|
> | Clipped HNN | 95.08 | 77.42 |
> | Clipped HNN + HypStructure | **95.19** | **78.05** |
>
> We observe that even with a classification loss function that optimizes for separability in the Hyperbolic space, or a Hyperbolic Neural Network as the backbone, our proposed methodology HypStructure is complementary and provides performance gains for better embedding the hierarchical relationship in the Hyperbolic space. (More details regarding the setup are in the global response above.)
>
> We will include the results with the experiments on hyperbolic losses and backbone in the revised version of the paper as well.
>
> [1] Hyperbolic Contrastive Learning for Visual Representations Beyond Objects, Ge et. al, CVPR ‘23
>
> ---
>
>
> ## Definition of dataset distance $d(D_i, D_j)$
>
> As we mention in line 75-76, the dataset distance $d(D_i, D_j)$ can vary depending on the choice of the distance metric and the design setup, hence we avoid writing an explicit definition in Sec. 2.1 and provide a general expression here instead. We follow this up with an example of the specific dataset distance for the l2-CPCC case in lines 97-98, where we see that $d(D_i, D_j) = \rho_{l2} =$ _the Euclidean distance of two averages of all feature vectors_ in each subset. On the other hand, for our proposed methodology HypStructure, you can either compute the metric $d(D_i, D_j)$ on Hyperbolic space, which first uses exponential map (Eq. 4) and then Klein average (Eq. 6) followed by the Poincare distance computation (line 180-181), or compute the means of the Euclidean features as in $\rho_{l2}$ and project the centroids to the Hyperbolic space using the exponential map, followed by the Poincare distance computation.
>
> ---
>
>
> ## Figure for Example 1 and 2
>
> The accompanying figures for both Example 1 and Example 2 are provided in Figure 1. More specifically, Example 1 only uses the tree nodes on the left part of Figure 1, which is what we refer to as the tree $\mathcal{T}$.
>
> The Example 2 uses both the left part of the Figure to describe nodes in the tree, C, D and E, and the right part of the Figure to visualize this corresponding setup in an attempt to embed the tree in the Euclidean space, based on the arguments described in the text in Example 2. We will make this more explicit in the paper description and the caption in the figure.
>
> ---
>
> ## Grammatical errors
>
> Thanks for the suggestions, we will proofread to improve readability and remove any grammatical errors in the camera ready version.
>
> ---
>
>
> ## Clarification for limitations
>
> The aforementioned comments misunderstand the limitations of our proposed method and we would like to provide a few clarifications regarding the limitations. First, the $l_2$-CPCC methodology suffers from the challenges of embedding a tree **exactly**, owing to the representational capacity of the Euclidean space which has zero curvature. This leads to high distortion when embedding a tree-like hierarchy using the $l_2$-CPCC, as also shown in the distortion metrics in Table 1. In contrast, Hyperbolic spaces allow for embedding tree-like data in finite dimensions with minimal distortion owing to the negative curvature of the spaces, and hence our proposed methodology - HypStructure does not suffer from the same limitation as the $l_2$-CPCC, and therefore reduces the distortion in embedding the hierarchy (Table 1).
>
> Additionally, while our submitted work experimented with primarily a Euclidean classification loss, (hence the corresponding statement in the limitations), over the rebuttal period based on the reviewers’ suggestions, we have experimented with other Hyperbolic losses and backbones, including the Hyperbolic SupCon loss and Clipped Hyperbolic Neural Network, which also shows improvements when combined with our proposed method, and hence our work is not limited to Euclidean classification losses. We will rephrase this statement to avoid further confusion.  We foresee a potential limitation - since our proposed method relies on the availability (or construction) of an external hierarchy for computing the HypCPCC objective, it can be challenging if the hierarchy is unavailable or noisy. We will include this in the limitations section.

---

> > ### Comment · Reviewer_gWWe · 2024-08-09
> >
> > Thanks for the authors' rebuttal. Personally, I think that the new results in the rebuttal have marginal improvements, and I am not sure if such improvements are statistically important, which decides the novelty of the proposed method. Therefore, I decide to keep my rate unchanged.

---

> > > ### Author Response · Authors · 2024-08-10
> > > **Response to concern about the statistical significance of the new results (Part 1)**
> > >
> > > Thanks for taking the time to go through the rebuttal, owing to the rebuttal timeline and limited space we were only able to report results corresponding to a single seed in our initial response, however we are now sharing a more exhaustive set of results with the hyperbolic settings with more evaluation metrics used in our paper, and you can find the results below (averaged over 3 seeds). **Additionally, to address your concerns about statistical significance, we performed the t-test for experiments with vs. without HypStructure, and highlighted the higher performance number only if the p-value is smaller than 0.05.**
> > >
> > > ## Experiments with the Hyperbolic Supervised Contrastive Loss
> > >
> > > Fine accuracies, coarse accuracies, and distortion measurements with the L_hyp loss .
> > >
> > > | Method | Dataset | $\delta_{rel}$ (lower) | CPCC (higher) | Fine Accuracy (higher)| Coarse Accuracy (higher)|
> > > |---|:---:|:---:|:---:|:---:|:---:|
> > > | L_hyp (HypSupCon)  | CIFAR10| 0.128 (0.007) | 0.745 (0.017) | 94.58 (0.04) | 98.96 (0.01)|
> > > | **L_hyp (HypSupCon) + HypStructure (Ours)**|CIFAR10 |**0.017 (0.001)**| **0.989 (0.001)** | **95.04 (0.02)** | **99.36 (0.02)**|
> > > | | | | | |
> > > | L_hyp (HypSupCon)  |CIFAR100|0.168 (0.002)|0.664 (0.012)|75.81 (0.06)|85.26 (0.07)|
> > > | **L_hyp (HypSupCon) + HypStructure (Ours)**|CIFAR100|**0.112 (0.005)**|**0.773 (0.008)**|**76.22 (0.14)**|**85.83 (0.06)**|
> > > | | | | | |
> > > | L_hyp (HypSupCon)  |ImageNet100|0.157 (0.004)|0.473 (0.004)|89.87 (0.01)|90.41 (0.01)|
> > > | **L_hyp (HypSupCon) + HypStructure (Ours)**|ImageNet100|**0.126 (0.002)** |**0.714 (0.003)**|**90.26 (0.01)**|**90.95 (0.01)**|
> > >
> > > Comparative Evaluation of OOD detection performance on a suite of OOD datasets with different ID datasets.
> > >
> > > OOD Detection AUROC Results on CIFAR10 (ID)
> > > | Method | SVHN | Textures| Places365 | LSUN | iSUN | Mean |
> > > |---|:---:|:---:|:---:|:---:|:---:|:---:|
> > > | L_hyp (HypSupCon)  |89.45 (0.18)|93.39 (0.24)|90.18 (0.09)|98.18 (0.19)	|91.31 (0.31)|92.502|
> > > | **L_hyp (HypSupCon) + HypStructure (Ours)**|**91.11 (0.21)**|**94.45 (0.13)**|**93.52 (0.33)**|**99.05 (0.17)**|**95.24 (0.42)**|**94.674**|
> > >
> > > OOD Detection AUROC Results on CIFAR100 (ID)
> > > | Method | SVHN | Textures| Places365 | LSUN | iSUN | Mean |
> > > |---|:---:|:---:|:---:|:---:|:---:|:---:|
> > > | L_hyp (HypSupCon)  |80.16 (0.08)|79.61 (0.26)|74.02 (0.45)|70.22 (0.18)|**82.35 (0.19)**|77.272|
> > > | **L_hyp (HypSupCon) + HypStructure (Ours)**|**82.28 (0.19)**|**83.51 (0.29)**|**77.95 (0.31)**|**86.64 (0.54)**|69.86 (0.87)|**80.048**|
> > >
> > > OOD Detection AUROC Results on ImageNet100 (ID)
> > >
> > > | Method | SUN | Places365| Textures | iNaturalist |Mean|
> > > |---|:---:|:---:|:---:|:---:|:---:|
> > > | L_hyp (HypSupCon)  |91.96 (0.18)|90.74 (0.26)|**97.42 (0.21)**	|94.04 (0.19)|93.54|
> > > | **L_hyp (HypSupCon) + HypStructure (Ours)**|**93.87 (0.05)**|**91.56 (0.13)**|97.04 (0.16)|**95.16 (0.24)**|**94.41**|
> > >
> > >
> > >
> > > ## Experiments using the Clipped Hyperbolic Neural Network Backbone
> > >
> > > Fine accuracies, coarse accuracies, and distortion measurements with the Clipped HNN backbone
> > >
> > > | Method | Dataset | $\delta_{rel}$ (lower) | CPCC (higher) | Fine Accuracy (higher)| Coarse Accuracy (higher)|
> > > |---|:---:|:---:|:---:|:---:|:---:|
> > > | Clipped HNN  | CIFAR10|0.084 (0.008)|0.604 (0.004)|94.81 (0.23)|89.71 (2.04)|
> > > | **Clipped HNN + HypStructure (Ours)**|CIFAR10 |**0.013 (0.002)**|**0.988 (0.001)**|94.97 (0.12)|**98.35 (0.22)**|
> > > | | | | | |
> > > | Clipped HNN   |CIFAR100|0.098 (0.001)|0.528 (0.009)|76.46 (0.26)|	49.26 (0.73)|
> > > | **Clipped HNN + HypStructure (Ours)**|CIFAR100|**0.064 (0.006)**|**0.624 (0.005)**|**77.96 (0.14)**|**55.46 (0.61)**|
> > > | | | | | |
> > >
> > >
> > >
> > > Comparative Evaluation of OOD detection performance on a suite of OOD datasets with different ID datasets.
> > >
> > > OOD Detection AUROC Results on CIFAR10 (ID)
> > >
> > > | Method | SVHN | Textures| Places365 | LSUN | iSUN | Mean |
> > > |---|:---:|:---:|:---:|:---:|:---:|:---:|
> > > | Clipped HNN   |92.63 (0.24)|90.74 (0.18)|88.46 (0.19)|95.66 (0.11)|92.41 (0.25)|91.98|
> > > | **Clipped HNN + HypStructure (Ours)**|**95.41 (0.44)**|**93.91 (0.32)**|**92.31 (0.41)**|**96.87 (0.21)**|**94.92 (0.31)**|**94.68**|
> > >
> > > OOD Detection AUROC Results on CIFAR100 (ID)
> > >
> > > | Method | SVHN | Textures| Places365 | LSUN | iSUN | Mean |
> > > |---|:---:|:---:|:---:|:---:|:---:|:---:|
> > > | Clipped HNN  |89.94 (0.16)|83.77 (0.23)|77.26 (0.33)|82.87 (0.29)|	82.35 (0.15)|83.23|
> > > |**Clipped HNN + HypStructure (Ours)**|**91.56 (0.21)**|**84.31 (0.09)**|**78.45 (0.28)**|**87.53 (0.52)**|**83.44 (0.37)**|**85.06**|
> > >
> > > ... (continued in the next response)

---

> > > ### Author Response · Authors · 2024-08-10
> > > **Response to concern about the statistical significance of the new results (Part 2)**
> > >
> > > ... (continued from Part 1 of this response)
> > >
> > > **Summary**: Based on the above results over multiple seeds, we can clearly observe that our proposed method leads to a **statistically significant** and consistent improvement in performance in terms of classification accuracies with a significantly more evident reduction in other metrics, including distortion of the representations, and a higher improvement on the mean AUROC in the OOD detection tasks, with improvements upto 3% across a suite of OOD datasets.
> > >
> > >
> > > >I think that the new results in the rebuttal have marginal improvements, and I am not sure if such improvements are statistically important, which decides the novelty of the proposed method
> > >
> > >
> > > We humbly disagree with the reviewers’ comment here and based on the above results, argue that the results would be a strong addition in demonstrating the wide-applicability of our proposed method **across tasks**, as has also been noted by Reviewer kf6R. Note that it has been discussed in multiple prior works [1] that improving accuracy on both classification as well as OOD detection tasks is fairly challenging.
> > >
> > > Furthermore, we would also like to point the reviewer to the global response where we clearly outline the novelty of our contribution which is not limited to the performance with hyperbolic losses/backbones, but is a lot more general, where **our work addresses the challenges of embedding explicit hierarchies and shows strong performance across tasks, learns interpretable representations and also draws important theoretical connections to tasks such as OOD detection, which are all very different from the prior works in the literature**.
> > >
> > > We hope that this clarifies the questions raised by the reviewer about the statistical significance and the novelty of the results. We would be happy to discuss any further questions about the work, and would really appreciate an increase in score if reviewers’ concerns are addressed.
> > >
> > >
> > > [1] OpenOOD: Benchmarking Generalized Out-of-Distribution Detection,Yang et. al,  NeurIPS ‘22, Datasets and Benchmarks Track
> > >
> > > [2] Learning Structured Representations by Embedding Class Hierarchy, Zeng et. al, ICLR ‘23

---

> > > > ### Author Response · Authors · 2024-08-13
> > > > **Regarding further discussion and any additional concerns**
> > > >
> > > > Dear Reviewer gWWe,
> > > >
> > > > We express gratitude for your time spent on reviewing our work and your valuable comments, and hope that
> > > > - the additional and exhaustive comparative experiments shared above help address your concerns about the statistical significance of our method's performance
> > > > - the positive comments and feedback from all three other reviewers (kf6R, 3c64 and dBDc) on our theoretical contributions, overall experimentation and insights may help clarify the novelty and impact of our work to the community
> > > >
> > > > With the rebuttal discussion window closing today, we look forward to engaging in further discussion to confirm if your concerns have been addressed and if there are any other questions regarding our work, that we can answer.

---

### Official Review · Reviewer_kf6R · 2024-07-12

**Soundness:** 3
**Presentation:** 3
**Contribution:** 3
**Rating:** 6
**Confidence:** 3

**Summary:**

This work introduces a regularization scheme based on Cophenetic Correlation Coefficient to more appropriately embed semantic label hierarchical structure in the representation. The method exploits the hierarchical benefits of hyperbolic space reformulating the CPCC regularization term to operate on the Poincare ball. The proposed method sees improvement in empirical performance demonstrating the effectiveness of the approach to learn a more separable embedding space for classification.

**Strengths:**

⁃	The authors present the work clearly with effective use of visual and writing structure. All figures/diagrams are useful in supporting the narrative and findings.

⁃	The method is simple, highly generalizable, and leads to improved performance on benchmark tasks. It can therefore, be seen as an advantageous tool in hyperbolic learning that could possibly lead to impact and adaptation by practitioners in the field.

⁃	The theoretical and analysis are generally good, with eigenspectrum analysis supporting your claims of hierarchical structure for the most part. This is a useful analysis that provides confidence in the findings supported by appropriate proofs.

⁃	Extensive details to support replication are provided.

**Weaknesses:**

⁃	From the visualizations presented of the embedding space, notably the UMAP, your embeddings seem to have collapsed to the boundary in many places limiting the inherent hierarchy of the embeddings, this results in a limited hierarchy being represented. This in turn, leads me to question the extent of hierarchies learnt, when discussing the core intention of the work, and the claims made. One would expect that greater performance could be achieved if this had been addressed. I am aware that boundary collapse is still an unsolved problem, but careful tuning can limit its effects.

⁃	The approach is simple but arguably not significantly novel given it is a hyperbolic reformulation of CPCC with minimal changes. With that being said, these simple methods do work somewhat well in practice and are useful to progressing the field.

⁃	The use of a Euclidean linear evaluation is a confusing direction. You are aiming to learn a hyperbolic embedding space that preserves hierarchy, yet for downstream tasks you employ a Euclidean classifier, why? You will lose the desirable properties you are aiming to capture.

⁃	Further experimentation on different hyperbolic models and downstream tasks would have helped demonstrate the generalization of the regularization to all of hyperbolic learning. Although, this cannot be expected in the rebuttal, it would have helped support the findings to present the work a more generalized approach.

**Questions:**

See weaknesses.

**Limitations:**

Broader impact statement and societal impact are seemingly missing from this work. These limitations including those limitations of the analysis should be highlighted in more detail, you mention the Euclidean loss as a limitation. If I have missed them however in the text, please correct me.

---

> ### Author Rebuttal · Authors · 2024-08-06
>
> We thank the reviewer for their detailed comments, and address each of the questions and weaknesses below.
>
> ## Boundary collapse
>
> Part of the design of our HypStructure methodology, **including the centering loss and embedding of internal nodes** mitigates boundary collapse.  Embedding the internal tree nodes in HypStructure - $T_{\text{int}}$ (as compared to only leaf nodes in prior work CPCC) and placing the root node at the center of the poincare disk with $l_{\text{center}}$ loss (discussed in lines 100-104, 186-194 in the main paper, lines 931-946 in the Appendix) embeds the hierarchy more accurately and the mitigate aforementioned issues. To demonstrate this intuition and evaluate its impact on the learnt hierarchy and representations, we first visualize the learnt representations from HypStructure, with and without these components - i.e. embedding internal nodes and a centering loss vs leaf only nodes, via UMAP in Figures 14 (CIFAR100) and 15 (ImageNet100) of the rebuttal PDF. We also provide a performance comparison (fine acc.) in the Table below
>
> |Method|CIFAR10|CIFAR100|ImageNet100|
> |-------|:-------------:|:-----:|:-----:|
> | HypStructure (leaf only) | 94.54 | 76.22  | 89.85  |
> | HypStructure (with internal nodes and centering) | **94.79**  | **76.68**  | **90.12** |
>
>
> First, based on Figures 14 and 15, one can note that in the leaf-only setting without embedding internal nodes + centering loss (figures on the left), the samples belonging to the fine classes which share the same parent (same color) are in close proximity, reflecting the hierarchy, however they are close to the boundary. With the embedding of internal nodes and a centering loss (right), we note that the representations are spread between the center (root) to the boundary as well as across the poincare disk, which is more representative of the original hierarchy. This also leads to performance improvements as can be seen in the table above.
>
> Note that we used the same $\beta = 0.01$ penalty parameter across datasets for convenience and easy reproducibility, and with a more careful tuning of $\beta$, the boundary collapse can be mitigated to a further extent. However, forcing the representations to be too close to the origin can lead to inter-class confusion and performance degradation as discussed in prior work [1] (Figure 4, page 7 of the paper).
>
> [1] Hyperbolic Busemann Learning with Ideal Prototype, Atigh et. al, NeurIPS ‘21
>
>
> ---
>
> ##  Novelty
>
> Thanks for your comments in support of the simplicity of our work. We would also like to refer the reviewer to our response regarding technical novelty in the global response above.
>
> ---
>
> ## Euclidean linear evaluation
>
> We provide **Euclidean linear evaluation on the Euclidean features**, as an added evaluation in the Appendix. Besides, it allows us to perform a fair comparison between important baselines, particularly l2-CPCC vs. HypStructure. To clarify, let us revisit our HypStructure pipeline: we first train an Euclidean loss SupCon which gives us an Euclidean feature (Fig. 6c), and then apply the exponential map to get the Poincare feature for CPCC computation. Because SupCon takes Euclidean inputs, it is reasonable to use Euclidean space for classification evaluation. From the visualizations in Figure 6b vs 6c, we note that the Hyperbolic regularizer empirically affects the geometry of the features even in the Euclidean space to be more structured and leads to performance improvements (Table 1, Fig. 7a, c)
>
> On the other hand, Hyperbolic classifiers are applicable if we change the backbone model. We experiment with a Hyperbolic model, Clipped HNN [1] and present the fine acc. results below (more details about the setup in the global response above), evaluated with Hyperbolic Multinomial Logistic Regression Layer in [2]:
>
> | Method | CIFAR10 | CIFAR100 |
> |---|:---:|:---:|
> | Clipped HNN | 95.08 | 77.42 |
> | Clipped HNN + HypStructure | **95.19** | **78.05** |
>
> We observe that HypStructure provides performance improvements with Hyperbolic backbones as well.
>
> [1] Clipped hyperbolic classifiers are super-hyperbolic classifiers, Guo. et al, CVPR ‘22
>
> [2] Hyperbolic neural networks, Ganea et. al, NeurIPS ‘18
>
> ---
>
> ## Further Experimentation
>
> We would also like to point the reviewer to additional experiments that we have performed with non-euclidean setups and we discuss more details about them in the global response.
>
> ---
>
> ## Broader impact and limitations
>
> Thanks for pointing this out, we would like to clarify that we experiment with the Euclidean loss primarily to provide a fair comparison with the prior works, and it is not a requirement (or limitation) for our HypStructure methodology. It can be combined with any other loss function (such as HypSupCon) or backbone (Clipped HNN) as we have demonstrated above, highlighting the wide applicability of our method.
>
> In addition to the discussion in Section 7, we enumerate other limitations which we foresee for future research and building on our methodology - our proposed method relies on the availability (or construction) of an external hierarchy for computing the HypCPCC objective, which might be challenging if the hierarchy is unavailable or noisy. In terms of Broader Impact, one potential area of impact that we recognize is the AI for science domain, where HypStructure can be helpful in learning more interpretable representations which are more reflective of the underlying relationships in the domain space. We include this in the revised manuscript.

---

> > ### Comment · Reviewer_kf6R · 2024-08-09
> > **Response to Rebuttal**
> >
> > Thank you for your insightful rebuttal, and clarification on points raised during the review.
> >
> > I'm grateful for your empirical evaluations and clarification on the boundary collapse and agree with the points made, however, I would still argue it would be beneficial to empirically demonstrate that collapse is not occurring given your strong claims made for hierarchy. Arguably, the small improvements when applying your centring method and accuracy metrics alone cannot refute the collapse claim or argue that the hierarchies you are aiming to capture are indeed captured. If I have misunderstood, or missed any further results that counter my points please do correct me.
> >
> > Thanks for the additional evaluation on hyperbolic networks this is a strong addition that addresses one of the identified weaknesses.
> >
> > I would emphasise the importance for a revised section outlining limitations, broader impact and societal for any revised manuscript, however, since you have identified these points in the rebuttal I have confidence the authors will provide this.
> >
> > Given my original review is positive, and that I still have a positive outlook for this work, I have maintained my score for now, and will revisit after further discussion with other reviewers.

---

> > > ### Author Response · Authors · 2024-08-10
> > > **Response to Reviewer kf6R**
> > >
> > > Thanks for the suggestion and for the support of our work! We will include the visualizations (Fig. 14 and 15)  and the new results on hyperbolic settings provided in the rebuttal in the camera ready version, along with a section outlining the limitations, and broader impact as well.

---

### Official Review · Reviewer_dBDc · 2024-07-12

**Soundness:** 3
**Presentation:** 3
**Contribution:** 2
**Rating:** 5
**Confidence:** 4

**Summary:**

The paper introduces HypStructure, a novel approach for learning structured representations using hyperbolic embeddings, which are well-suited for modeling hierarchical relationships due to their tree-like structure. The method incorporates a hyperbolic tree-based representation loss and a centering loss to embed label hierarchies into feature spaces with minimal distortion.

Experiments demonstrate HypStructure's effectiveness in improving classification accuracy, especially in low-dimensional scenarios, and enhancing Out-of-Distribution (OOD) detection performance without compromising in-distribution accuracy.

**Strengths:**

1. Although it is already theoretically proved in the related work of [70, 72] that it is not possible to embed a tree in Euclidean space without loss, it is still informative to see that In section 2.2, example 1 and example 2 give good counter-examples to show this property.

2. The paper is well-written and easy to follow, and the proposed model is simple yet effective.

3. The paper provides a formal eigenvalue analysis that links the geometry of the learned representations to their performance

**Weaknesses:**

1. Sections 2.1, 2.2, and 3.1 are all from existing literature, which limited the contribution of the paper. Although the operations described in Section 3.1 are common hyperbolic operations, this section still lacks proper reference to the related papers.

2. In HypCPCC, the authors proposed two alternatives of the loss,
    * Map Euclidean vectors to Poincaré space then average.
    * Averaging the Euclidean vectors then map to Poincaré space.
In the 1st alternative, The use of Klein weighted average incurs extra computation, Is it worth doing so?, In the 2nd alternative is exactly the same as [r2], which also calculates the prototypes for each class in hyperbolic space and then map to Poincaré space, [r2] also deploys supervised constructive learning, but the reference is missing and comparison is not stated.

3. The statement in Theorem 5.1 is incorrect, an entry of $K$, denoted as $r^h$ should be a vector, but the theorem stated that $\lambda_0 = 1 - r^1$, which does not make sense.

3. Incorrect (but fixable) definition in line 708, the proof used $\| u \| = \| v \| = 1-\epsilon $ but in the proof the authors used the fact that $\| u \| = \| v \| = 1-\epsilon^2 $

4. Incorrect proof in Corollary A.1, the last row of the proof does not hold, Poincaré distance cannot be the same as Euclidean distance, "growing in the same trend" does not mean "proportional to".

[r2] Long, Teng, et al. "Searching for actions on the hyperbole." CVPR2020

**Questions:**

1. What is the rationale behind choosing the Klein weighted average for the first alternative in HypCPCC, considering the extra computation it incurs?
2. Can the authors provide a comparison to [r2], which is (a special case of) their second alternative of HypCPCC?

**Limitations:**

The authors did not discuss the limitations in the paper.

---

> ### Author Rebuttal · Authors · 2024-08-06
>
> We thank the reviewer for detailed comments, and address each of the questions and weaknesses below.
>
> ## Contribution
>
> While our work builds on prior research, this does not diminish the contribution of our work and we request the reviewer to kindly refer to our note on novelty in the global response, we clearly outline the key differences between our work and prior works.
>
> ---
>
> ## Reference to Hyperbolic ops.
>
> Thanks for pointing this out, we will include references to common Hyperbolic operations in the revised manuscript [1]
>
> [1] Hyperbolic trigonometry and its applications in the Poincare ball model of Hyperbolic geometry, Ungar A., Computers & Mathematics with Applications, 2001
>
> ---
>
> ## Comparison to [r2]
>
> Thanks for pointing out this work, indeed [r2] is a related work, we will cite it in our revised manuscript and have attempted to provide a comparison below. However, first we discuss the key differences between [r2] and our work.
>
> 1. **The usage of hierarchy and “prototypes” differ significantly.** [r2] is a two-step method, where first an action *word* Hyperbolic embedding is created based on action hierarchies (each word is a single tree node), and second a video embedding is learnt using positions of action words (called prototypes in [r2]) in the Hyperbolic space. In contrast, our method learns the hierarchical image embeddings end-to-end, does not depend on word embeddings, and performs a centroid computation on the label hierarchy tree nodes (many images belong to a single label node, hence class prototypes are computed by averaging).
>
> 2. **The nature of components to learn hierarchical embeddings are different**. In [r2], both losses $L_H$ (Eq. 2 in [r2], as in [2]) and $L_2$ (Eq. 6 in [r2], as in [3]) are ranking losses, whereas HypCPCC factors in absolute values of the node-to-node distances. The learned hierarchy with HypCPCC will not only (implicitly) encode the correct parent-child relation like [2, 3], but also learn more complex and weighted hierarchical relationships more accurately (Fig. 16 in the rebuttal PDF).
>
> We attempt to compare HypStructure and [r2] now. Since losses $L_H$ and $L_2$ have similar motivation as HypStructure, we add them as regularizers to the Hyperbolic classification backbone Clipped HNN and compare with our method:
>
> |Method|CIFAR10|CIFAR100|
> |------|:-----:|:------:|
> |Clipped HNN|95.08|77.42|
> |Clipped HNN + $L_H$|94.78|77.44|
> |Clipped HNN + HypStructure|**95.19**|**78.05**|
>
> We see HypStructure achieves better performance than $L_H$. We face NaN issues while training with L_2 and other losses, and discuss the setup details in the global response.
>
> [2] Poincaré embeddings for learning hierarchical representations, Nickel et. al, NIPS ‘17
>
> [3] Hyperbolic entailment cones for learning hierarchical embeddings, Ganea et. al, ICML ‘18
>
> ---
>
> ## Rationale behind Klein avg.
>
> We experiment with both HypCPCC variants mentioned in line 177-185. The 2nd variant, however, given our understanding, is not a special case of [r2], since it still requires a Euclidean averaging step for class prototype computation. We empirically observe performance improvements in accuracy using the 1st variant across datasets below (fine acc.).
>
> |Method|CIFAR10|CIFAR100|ImageNet100|
> |------|-------|--------|-----------|
> |Euc. (2)|94.56|75.64|90.08|
> |Klein (1)|**94.79**|**76.68**|**90.12**|
>
> ---
>
> ## Theorem 5.1 entry of K
>
> Since $K$ is $\mathbb{R}^{n\times n}$ (line 289), an *entry* of $K$ is -  $K_{i,j}$, the $i$-th row and $j$-th column scalar value of K. Each entry of $K$ can be denoted as $r^h$, where $r^h$ is the lowest common ancestor (LCA) of two samples represented by the $i$-th row and $j$-th column respectively so it is indeed a scalar. We also refer the reviewer to the matrix in lines 714-715, where each entry of $K$, denoted as $r^h_{i,j}$, is a scalar value.  (i,j subscript can be ignored, see 718-719, since every two leaves sharing the LCA of the same height have the same tree distance). We will make this more explicit in the revised version to avoid confusion.
>
> ---
>
> ## “The proof used $|𝑢|=|𝑣|=1−𝜖$…”
> Thanks for pointing out this typo due to our cluttered notation! We intended to mean $|u|^2 = 1 - \text{(a small number)}$, however the conclusion of the proof still holds with this change, as the reviewer noted. Let us rewrite the proof: in practice with $clip^1$ projection, to be consistent with the notation in the main body, we set $|u|=|v| = 1 - \epsilon$ where $\epsilon$ is a very small number. Then $|u|^2 = (1 - \epsilon)^2 = 1 - 2\epsilon + \epsilon^2$. Since $\epsilon$ is very small, we define $2\epsilon - \epsilon^2$ as $\xi$, making $|u|^2 := 1 - \xi$ where $\xi$ is still a small number. Then replacing all $\epsilon$ with $\xi$ in line 708-709, we get the same conclusion. We will correct this error in the revised version.
>
> ---
>
> ## “Incorrect proof in Corollary A.1...”
>
> Apologies for the confusion. With the statement “growing in the same trend”, we meant that Poincaré(u,v) is monotonically increasing with Euclidean(u,v), and the proof after Corollary A.1 will not depend on the metric, as can be seen from comments in 710. This is true since with $\text{clip}^1$ transformation, Poincaré is approximately the log-scalar transform of Euclidean distance. The monotonically increasing property ensures the relative order of any two entries for Euclidean CPCC and Poincare CPCC matrices in K to be the same. Then, we can argue about the structure of K, either Euclidean or Poincare, to have the hierarchical diagonalized structure as in Fig 8a. But indeed technically, due to the logarithm, this is not “proportional to”. We will remove this notation and make it more clear.
>
> ---
>
> ## Limitations
>
> Another potential limitation - our method relies on the availability (or construction) of an external hierarchy for computing the HypCPCC objective, which might be challenging if the hierarchy is unavailable or noisy. We will include this in the next iteration of the paper.

---

> > ### Comment · Reviewer_dBDc · 2024-08-13
> >
> > Thank you for your detailed response and the clarifications provided. I appreciate the effort you put into addressing my concerns. After reviewing your explanation, most of my concerns have been resolved.
> >
> > Although I acknowledge that the novelty and technical contribution might be somewhat limited, as pointed out by Reviewer 3c64, I believe that your proofs and the extensive experiments you conducted will still provide value to the community. In light of this, I will be raising my rating.

---

> > > ### Author Response · Authors · 2024-08-13
> > > **Response to Reviewer dBDc**
> > >
> > > Dear Reviewer dBDc,
> > >
> > > Thanks for your valuable feedback in the support of our work and for raising your rating! We are happy to see that our responses could address your concerns. We will include the clarifications and corrections to the proofs based on your suggestions, in the revised manuscript, and if you have any further questions for our work, we would be happy to continue the discussion.

---

### Author Rebuttal · Authors · 2024-08-06

We sincerely thank the reviewers for your valuable reviews and constructive feedback that has helped us improve our work. We first share the results of additional experiments we conducted based on these suggestions which demonstrate the wide applicability of our method, and then summarize our technical contributions and its novelty.

## Additional experiments

1. **Hyperbolic Backbones and Losses**

According to the suggestions from reviewers dBDC, kf6R, and gWWe, we experimented with a Hyperbolic backbone, Clipped HNN [1], which is developed on the Hyperbolic MLR layer in the work Hyperbolic Neural Networks [2]. We also experiment with a composite objective combining this backbone and our HypStructure regularizer.

For comparison, we also experiment with the objective function in the seminal work Poincaré Embedding [3] as a regularizer, since the methodology also aims to learn Hyperbolic (word) embeddings (as suggested by reviewer dBDc for learning action embeddings in [r2]). Following the setup in our paper, we use the first alternative for both HypStructure and PoincaréEmb in this comparison, i.e., we first ensure the features are in Poincaré ball, followed by the Klein averaging for each class node, and apply the regularizer accordingly. The fine accuracy results are as follows:

|Method|CIFAR10|CIFAR100|
|---|:---:|:---:|
|Clipped HNN|95.08|77.42|
|Clipped HNN+PoincareEmb|94.78|77.44|
|Clipped HNN+HypStructure|**95.19**|**78.05**|

We observe that HypStructure is a flexible regularizer compatible with Hyperbolic backbones and it improves the fine-level classification accuracy from the Clipped HNN baseline, and performs better than Clipped HNN + Poincare Embedding [3] as well. We also experimented with the Poincare Embedding [3] regularizer with the Supervised Contrastive (SupCon) loss, however the training runs into NaN issues due to numerical instability in the PoincareEmb computation. Note that for this experiment, we could not provide results on ImageNet100, since in our main experiments in Table 1 for ImageNet100, we rely on fine-tuning a pre-trained ResNet backbone on ImageNet-1K, whereas we were unable to find a comparable pretrained Hyperbolic model for a fair comparison to results in Table 1.

We also replaced the Euclidean Supervised Contrastive Loss, with the **Hyperbolic Supervised Contrastive Loss** $L_{\text{hyp}}$ as proposed in [4], and experiment with HypStructure in combination with this Hyperbolic loss. We report the results on different datasets below (fine acc.)

|Method|CIFAR10|CIFAR100|ImageNet100|
|---|:---:|:---:|:---:|
|L_hyp (HypSupCon)|94.64|75.77|90.02|
|L_hyp (HypSupCon) + HypStructure (Ours)|**95.06**|**76.08**|**90.31**|

We observe that HypStructure works well with Hyperbolic contrastive losses as well, and in fact leads to improved absolute accuracies on CIFAR10 and ImageNet100 than our previous results using the supervised contrastive losses.

[1] Clipped hyperbolic classifiers are super-hyperbolic classifiers, Guo. et al, CVPR ‘22

[2] Hyperbolic neural networks, Ganea et. al, NeurIPS ‘18

[3] Poincaré embeddings for learning hierarchical representations, Nickel et. al, NeurIPS ‘17

[4] Hyperbolic Contrastive Learning for Visual Representations Beyond Objects, Ge et. al, CVPR ‘23

[r2] Searching for actions on the hyperbole, Long et. al, CVPR ‘20



2. **Understanding the components of HypStructure**

Based on the suggestion from reviewers dBDC, kf6R, we perform additional experiments and visualizations to understand the working of HypStructure, specifically:


- 2a. **The role of centering loss for the root and embedding internal nodes** - see Table below and comparison of visualizations in Figures 14 and 15 in the PDF

 |Method|CIFAR10|CIFAR100|ImageNet100|
 |---|:---:|:---:|:---:|
|HypStructure (leaf only)|94.54|76.22|89.85|
|HypStructure (with internal nodes and centering)|**94.79**|**76.68**|**90.12**|

- 2b. **The role of Klein averaging vs Euclidean averaging** for improved empirical performance

|Method|CIFAR10|CIFAR100|ImageNet100|
|------|-------|--------|-----------|
|Euc. (2)|94.56|75.64|90.08|
|Klein (1)|**94.79**|**76.68**|**90.12**|

- 2c. **The capacity of HypStructure to learn complex hierarchical relationships** with differently weighted trees - Figure 16 in the PDF.

---

## Novelty & Contribution

1. **Complementary nature to prior Hyperbolic works and address limitations of l2-CPCC**: While our work draws inspiration from recent works in Hyperbolic machine learning, we propose a novel, practical and effective framework that addresses the limitations of the l2-CPCC,as we clearly demonstrate with the strong empirical performance on large-scale datasets and visualization of learned embeddings. Furthermore, our approach offers a flexible framework that can be complementary and used in conjunction with other Hyperbolic losses/backbones for performance gains. We want to highlight that our work is very different from the majority of prior works in Hyperbolic learning that assume a latent implicit hierarchy in the learning process, while our work solves the problem of leveraging explicit external hierarchical knowledge in representation learning.

2. **Provable structured representation learning**: To the best of our knowledge, our work is one of the first to draw theoretical connections between the paradigm of hierarchy-informed representation learning and OOD detection. These insights can be very useful in designing theoretically grounded, efficient and accurate representation learning methods that can learn general representations useful across tasks in practice as we have demonstrated for classification and OOD detection, while maintaining interpretable representations.

We believe our work makes several novel contributions, and we anticipate that our work will inspire further research on embedding structured information and usage of Hyperbolic geometry. We hope this adequately justifies our contributions to the reviewers.

---

> ### Author Response · Authors · 2024-08-10
> **Additional Results for the experiments with Hyperbolic Settings**
>
> We are now sharing a more exhaustive set of results with the hyperbolic settings with more evaluation metrics used in our paper, with the results below (averaged over 3 seeds). **We also perform the t-test for experiments with vs. without HypStructure, and highlighted the higher performance number only if the p-value is smaller than 0.05, for statistical significance**
>
> ## Experiments with the Hyperbolic Supervised Contrastive Loss
>
> Fine accuracies, coarse accuracies, and distortion measurements with the L_hyp loss .
>
> | Method | Dataset | $\delta_{rel}$ (lower) | CPCC (higher) | Fine Accuracy (higher)| Coarse Accuracy (higher)|
> |---|:---:|:---:|:---:|:---:|:---:|
> | L_hyp (HypSupCon)  | CIFAR10| 0.128 (0.007) | 0.745 (0.017) | 94.58 (0.04) | 98.96 (0.01)|
> | **L_hyp (HypSupCon) + HypStructure (Ours)**|CIFAR10 |**0.017 (0.001)**| **0.989 (0.001)** | **95.04 (0.02)** | **99.36 (0.02)**|
> | | | | | |
> | L_hyp (HypSupCon)  |CIFAR100|0.168 (0.002)|0.664 (0.012)|75.81 (0.06)|85.26 (0.07)|
> | **L_hyp (HypSupCon) + HypStructure (Ours)**|CIFAR100|**0.112 (0.005)**|**0.773 (0.008)**|**76.22 (0.14)**|**85.83 (0.06)**|
> | | | | | |
> | L_hyp (HypSupCon)  |ImageNet100|0.157 (0.004)|0.473 (0.004)|89.87 (0.01)|90.41 (0.01)|
> | **L_hyp (HypSupCon) + HypStructure (Ours)**|ImageNet100|**0.126 (0.002)** |**0.714 (0.003)**|**90.26 (0.01)**|**90.95 (0.01)**|
>
> Comparative Evaluation of OOD detection performance on a suite of OOD datasets with different ID datasets.
> OOD Detection AUROC Results on CIFAR10 (ID)
> | Method | SVHN | Textures| Places365 | LSUN | iSUN | Mean |
> |---|:---:|:---:|:---:|:---:|:---:|:---:|
> | L_hyp (HypSupCon)  |89.45 (0.18)|93.39 (0.24)|90.18 (0.09)|98.18 (0.19)	|91.31 (0.31)|92.502|
> | **L_hyp (HypSupCon) + HypStructure (Ours)**|**91.11 (0.21)**|**94.45 (0.13)**|**93.52 (0.33)**|**99.05 (0.17)**|**95.24 (0.42)**|**94.674**|
>
> OOD Detection AUROC Results on CIFAR100 (ID)
> | Method | SVHN | Textures| Places365 | LSUN | iSUN | Mean |
> |---|:---:|:---:|:---:|:---:|:---:|:---:|
> | L_hyp (HypSupCon)  |80.16 (0.08)|79.61 (0.26)|74.02 (0.45)|70.22 (0.18)|**82.35 (0.19)**|77.272|
> | **L_hyp (HypSupCon) + HypStructure (Ours)**|**82.28 (0.19)**|**83.51 (0.29)**|**77.95 (0.31)**|**86.64 (0.54)**|69.86 (0.87)|**80.048**|
>
> OOD Detection AUROC Results on ImageNet100 (ID)
>
> | Method | SUN | Places365| Textures | iNaturalist |Mean|
> |---|:---:|:---:|:---:|:---:|:---:|
> | L_hyp (HypSupCon)  |91.96 (0.18)|90.74 (0.26)|**97.42 (0.21)**	|94.04 (0.19)|93.54|
> | **L_hyp (HypSupCon) + HypStructure (Ours)**|**93.87 (0.05)**|**91.56 (0.13)**|97.04 (0.16)|**95.16 (0.24)**|**94.41**|
>
>
>
> ## Experiments using the Clipped Hyperbolic Neural Network Backbone
>
> Fine accuracies, coarse accuracies, and distortion measurements with the Clipped HNN backbone
>
> | Method | Dataset | $\delta_{rel}$ (lower) | CPCC (higher) | Fine Accuracy (higher)| Coarse Accuracy (higher)|
> |---|:---:|:---:|:---:|:---:|:---:|
> | Clipped HNN  | CIFAR10|0.084 (0.008)|0.604 (0.004)|94.81 (0.23)|89.71 (2.04)|
> | **Clipped HNN + HypStructure (Ours)**|CIFAR10 |**0.013 (0.002)**|**0.988 (0.001)**|94.97 (0.12)|**98.35 (0.22)**|
> | | | | | |
> | Clipped HNN   |CIFAR100|0.098 (0.001)|0.528 (0.009)|76.46 (0.26)|	49.26 (0.73)|
> | **Clipped HNN + HypStructure (Ours)**|CIFAR100|**0.064 (0.006)**|**0.624 (0.005)**|**77.96 (0.14)**|**55.46 (0.61)**|
> | | | | | |
>
>
>
> Comparative Evaluation of OOD detection performance on a suite of OOD datasets with different ID datasets.
> OOD Detection AUROC Results on CIFAR10 (ID)
>
> | Method | SVHN | Textures| Places365 | LSUN | iSUN | Mean |
> |---|:---:|:---:|:---:|:---:|:---:|:---:|
> | Clipped HNN   |92.63 (0.24)|90.74 (0.18)|88.46 (0.19)|95.66 (0.11)|92.41 (0.25)|91.98|
> | **Clipped HNN + HypStructure (Ours)**|**95.41 (0.44)**|**93.91 (0.32)**|**92.31 (0.41)**|**96.87 (0.21)**|**94.92 (0.31)**|**94.68**|
>
> OOD Detection AUROC Results on CIFAR100 (ID)
>
> | Method | SVHN | Textures| Places365 | LSUN | iSUN | Mean |
> |---|:---:|:---:|:---:|:---:|:---:|:---:|
> | Clipped HNN  |89.94 (0.16)|83.77 (0.23)|77.26 (0.33)|82.87 (0.29)|	82.35 (0.15)|83.23|
> |**Clipped HNN + HypStructure (Ours)**|**91.56 (0.21)**|**84.31 (0.09)**|**78.45 (0.28)**|**87.53 (0.52)**|**83.44 (0.37)**|**85.06**|
>
> **Summary**: Based on the above results over multiple seeds, we can clearly observe that our proposed method leads to a **statistically significant** and consistent improvement in performance in terms of classification accuracies with a significantly more evident reduction in other metrics, including distortion of the representations, and a higher improvement on the mean AUROC in the OOD detection tasks, with improvements upto 3% across a suite of OOD datasets.

---

### Decision · Program_Chairs · 2024-09-25

**Decision:**

Accept (poster)

**Comment:**

This paper has received active consideration and discussion by the reviewers, which in turn changed the scoring of the paper towards a consensus. The reviewers appreciate the theoretical nature of the paper and find the idea simple yet clear and effective. There were a number of limitations that required a rebuttal. The AC considers the questions about additional baselines, analyses, and issues with the theory solved. The reviewers have also reached a consensus towards accept, albeit close the borderline. The AC recommends acceptance since the theory and the methodology of the paper are of interest to the community and the additional experiments and clarifications of the rebuttal strengthen the paper. A number of initial reviewer concerns are a result of a lack of referencing; despite citing >100 papers, many papers on hyperbolic learning for computer vision specifically were missed. The AC urges the authors to includes the outcomes of the rebuttal in the paper and to include both the references mentioned by the reviewers and other highly related works (especially visual learning with hyperbolic hierarchical embeddings).